# TRAINING-TIME NEURON ALIGNMENT FOR IMPROVING LINEAR MODE CONNECTIVITY AND MODEL FUSION

## ABSTRACT

In deep learning, Stochastic Gradient Descent (SGD) will find different solutions that are functionally similar but far away from each other in the parameter space. The loss landscape of linearly connecting two SGD solutions is called Linear Mode Connectivity (LMC), which often shows barriers. Current neuron alignment methods seek to find a network permutation that can map two SGD solutions into the same loss basin to improve LMC and model fusion. However, these methods are post-hoc and usually require large computations due to the astronomical number of permutation matrices. Can we realize training-time neuron alignment? In this paper, we first hypothesize that it can be realized by learning into an effective *subspace*. First, we provide a preliminary theoretical result to support the hypothesis. We further propose a subspace algorithm for partially fixing neuron weights to reduce the potential permutation symmetries without hurting accuracy. It is found that by applying our training-time alignment method, the LMC is largely improved and the required computation for post-matching is reduced. Interestingly, we also find random pruning at initialization can improve connectivity, which validates our subspace hypothesis. Lastly, we propose two algorithms, incorporating training-time neuron alignment in federated learning, to showcase its prospects in boosting model fusion even under heterogeneous datasets.

## 1 INTRODUCTION

Understanding the loss landscape of Deep Neural Networks (DNNs) is the key to understanding the mechanisms and training dynamics behind generalization and optimization (Li et al., 2018b; Fort & Jastrzebski, 2019; Simsek et al., 2021; Vlaar & Frankle, 2022), and it is still an open problem. Empirical findings demonstrate that Stochastic Gradient Descent (SGD) will find many minima that are functionally similar but far away from each other in parameter space (Draxler et al., 2018; Zhang et al., 2021; Entezari et al., 2022). The literature in Linear Mode Connectivity (LMC) suggests that if we linearly interpolate two independently trained networks, which have the same initialization and trainset but have different SGD random seeds (i.e., batch orders), there will be a loss barrier in the landscape (Draxler et al., 2018; Garipov et al., 2018; Ainsworth et al., 2022) (also see Figure 3). The loss barrier reflects that the two SGD solutions fall into two different loss basins that cannot be linearly connected, and it is detrimental to model fusion (Ainsworth et al., 2022; Li et al., 2022).

The reasons behind the barrier in LMC are mainly the over-parameterization and permutation invariance (also known as permutation symmetry) properties of DNNs. Over-parameterization explains why there are abundant minima found by SGD (Zhang et al., 2021; Neyshabur et al., 2017; Safran & Shamir, 2018). Permutation invariance suggests that the function of the network can remain the same while changing the permutations of neurons, which can result in many functionally same but geometrically different solutions (Ainsworth et al., 2022; Wang et al., 2020a; Tatro et al., 2020). Previous works seek to achieve neuron alignment for improving LMC by utilizing the permutation invariance property. In Entezari et al. (2022), it is conjectured that if taking all permutations into account, all SGD solutions can be mapped into the same loss basin where no barrier in LMC. Git Re-Basin (Ainsworth et al., 2022) further validates this conjecture by proposing three algorithms to find such permutations.

However, post-hoc neuron alignment is a hard combinatorial optimization problem. As stated in Ainsworth et al. (2022), even for a three-layer MLP with 512 widths, the number of permutation symmetries is nearly 10 ^ 3498. It is very computationally expensive and difficult to find an appropriate permutation to align one network with the other. For the scenarios where alignment among multiple models is needed, especially federated learning, the post-hoc alignment task is more challenging (Wang et al., 2020b).

Instead of the post-matching, in this paper, we provide a new perspective from the *training time*. We explore *the potential of training-time neuron alignment*, which can improve LMC and also reduce the burdens of post-hoc alignment after training.

To realize training-time alignment, we first delve into the causes behind the LMC barrier. Over-parameterization and permutation invariance imply that the optimization space of training is so large that two independently trained models can hardly converge to the same loss basin. Intuitively, if we can map the parameters of models into a consistent and effective subspace[1] during training, so the neurons will be more aligned. The "effective" means that i) learning in this subspace will not hurt the accuracies of the trained models; ii) the barrier in LMC can be reduced; and iii) the convex fusion of multiple models can be improved. Towards this goal, we have the following **contributions**.

- We discover the neuron alignment problem from the perspective of training time, which provides new insights. We hypothesize that learning in *subspaces* can reach better neuron alignment.
- We make preliminary verification of the hypothesis theoretically. Further, we propose an algorithm for the effective subspaces, validated under LMC and model fusion.
- We extend the training-time alignment method of LMC in federated learning, where model fusion from multiple heterogeneous sources is required. It is shown that our two methods can have dominant empirical improvements by making neurons more aligned during training.

## 2 BACKGROUND

In this section, we provide the basic backgrounds and definitions regarding linear mode connectivity and permutation invariance. Additionally, the preliminary of federated learning is in Appendix D.

**Linear mode connectivity (LMC).** In this paper, we focus on the linear mode connectivity of two SGD solutions, which have the same initialization but different data orders[2]. We present the definitions of loss barrier and accuracy barrier below.

**Definition 2.1** *(**Loss barrier** (Entezari et al., 2022)) Let $f_{\mathbf{w}}(\cdot)$ be a function represented by a neural network with parameter vector $\mathbf{w}$ that includes all parameters and $\mathcal{L}(\mathbf{w})$ be the any given loss (e.g., train or test error) of $f_{\mathbf{w}}(\cdot)$. Given two independently trained networks $\mathbf{w}_1$ and $\mathbf{w}_2$, let $\mathcal{L}(\alpha\mathbf{w}_1 + (1-\alpha)\mathbf{w}_2)$, for $\alpha \in [0,1]$ be the loss of the linearly interpolated network. The loss barrier $B_{loss}(\mathbf{w}_1, \mathbf{w}_2)$ along the linear path between $\mathbf{w}_1$ and $\mathbf{w}_2$ is defined as the highest difference between the loss of the interpolated network and linear interpolation of the loss values of the two networks:*

$$B_{loss}(\mathbf{w}_1, \mathbf{w}_2) = \sup_{\alpha}[\mathcal{L}(\alpha\mathbf{w}_1 + (1-\alpha)\mathbf{w}_2)] - [\alpha\mathcal{L}(\mathbf{w}_1) + (1-\alpha)\mathcal{L}(\mathbf{w}_2)]. \tag{1}$$

The loss barrier of the above definition is not bounded. To better depict and compare the barrier changes, we then propose a definition of the accuracy barrier which is bounded within $[0, 1]$.

**Definition 2.2** *(**Accuracy barrier**) Let $\mathcal{A}(\mathbf{w})$ be the accuracy (e.g., train or test accuracy) of $f_{\mathbf{w}}(\cdot)$. Let $\mathcal{A}(\alpha\mathbf{w}_1 + (1-\alpha)\mathbf{w}_2)$, for $\alpha \in [0,1]$ be the accuracy of the linearly interpolated network. The accuracy barrier $B_{acc}(\mathbf{w}_1, \mathbf{w}_2)$ along the linear path between $\mathbf{w}_1$ and $\mathbf{w}_2$ is defined as the highest ratio of the interpolated network's accuracy drop to the averaged accuracy:*

$$B_{acc}(\mathbf{w}_1, \mathbf{w}_2) = \sup_{\alpha}\left[1 - \frac{\mathcal{A}(\alpha\mathbf{w}_1 + (1-\alpha)\mathbf{w}_2)}{\alpha\mathcal{A}(\mathbf{w}_1) + (1-\alpha)\mathcal{A}(\mathbf{w}_2)}\right]. \tag{2}$$

The above definition maps the barrier into $[0, 1]$. If the accuracy barrier is 0, it means no barrier exists along the linear interpolation path; else if the barrier is nearly 1, it means the generalization of the interpolated model is nearly zero, and its prediction is no better than random guessing.

**Permutation invariance.** Permutation invariance refers to the property that the positions (i.e., permutations) of neurons of a given network can be changed without changing the network's function, and it is also known as permutation symmetry (Ainsworth et al., 2022). We take a multi-layer MLP as an example to demonstrate the property.

---

[1]The word "subspace" in this paper means: given a network, making the learnable parameters/degrees of freedom reduced. We note that there may exist some other subspace definitions/meanings, but they are not under this paper's scope.

[2]We note that there are other forms of LMC, such as the LMC from the initialization and the trained model (Vlaar & Frankle, 2022), and the LMC between two models with different initializations (Entezari et al., 2022). While in this paper, we focus on LMC and model fusion with specific applications in federated learning, we only consider the LMC cases with the same initialization.

Assume an MLP network has $L+1$ layers (containing input and output layer), and each layer contains $J_l$ neurons, where $l \in \{0, 1, \cdots, L\}$ is the layer index. $J_0$ and $J_L$ are input and output dimensions. We denote the parameters of each layer as the weight matrix $W_l \in \mathbb{R}^{J_l \times J_{l-1}}$ and the bias vector $b_l \in \mathbb{R}^{J_l}$, $l \in \{1, 2, \cdots, L\}$. The input layer does not have parameters. We use $h_l \in \mathbb{R}^{J_l}$ as the outputs of the $l$-th layer. We have $h_l = \sigma_l(W_l h_{l-1} + b_l)$, where $\sigma_l(\cdot)$ is the element-wise activation function, e.g., ReLU. We use $\Pi \in \{0, 1\}^{J \times J}$ as a permutation matrix that satisfies $\sum_j \Pi_{\cdot, j} = 1$ and $\sum_j \Pi_{j, \cdot} = 1$. By applying the permutation matrices to the layers, the network function remains unchanged. For the $l$-th layer, the layer-wise permutation process is

$$h_l = f_l(\Pi_l W_l \Pi_{l-1}^T h_{l-1} + \Pi_l b_l), \tag{3}$$

where $\Pi_0 = I$ and $\Pi_L = I$, meaning that the input and output are not shuffled. We note that the permutation matrices have the following properties:

$$\Pi^T \Pi = I, \Pi\mathbf{a} + \Pi\mathbf{b} = \Pi(\mathbf{a} + \mathbf{b}), \Pi\mathbf{a} \odot \Pi\mathbf{a} = \Pi(\mathbf{a} \odot \mathbf{b}), \sigma(\Pi\mathbf{x}) = \Pi\sigma(\mathbf{x}), \tag{4}$$

where $I$ is the identity matrix, $\odot$ denotes Hadamard product, and $\sigma(\cdot)$ is an element-wise function. **The connection between LMC and permutation invariance.** In Entezari et al. (2022); Ainsworth et al. (2022), it is conjectured that if applying appropriate permutation matrices to the networks, two SGD solutions that have barriers before can be mapped into the same loss basin and linearly connected (with low barriers). In previous literature, post-hoc matching methods are proposed to approximate the right permutations (Ainsworth et al., 2022; Peña et al., 2023). However, the number of permutation matrices is astronomical, and finding such appropriate permutations is hard. In this paper, instead of post-matching, we provide a new perspective by exploring the potential of improving linear mode connectivity in *training time*.

## 3 Hypothesis and Preliminary Theoretical Analysis

Due to the numerous parameters and permutation symmetries, during training, SGD will find solutions that are far from each other in the landscapes. Therefore, we make the following hypothesis.

**Hypothesis 3.1** *If we can reduce the potential of permutation symmetries by learning the models in a unified subspace, the linear mode connectivity will be improved.*

For the subspace hypothesized in Hypothesis 3.1, the number of learned parameters is reduced and the neurons' updates are regularized toward a more unified direction, as a result, the final trained models will be more connected in the parameter geometry.

We first make a preliminary theoretical analysis of the subspace hypothesis, shown in Theorem 3.2 (proof is in Appendix B). The main idea is to treat the linear interpolated landscape of the barrier as a function of parameter $\alpha$, and the connectivity can be depicted by the first and second derivatives of the function.

**Theorem 3.2** *We define a two-layer neural network with ReLU activation, and the function is $f_{\boldsymbol{v}, \boldsymbol{U}}(\boldsymbol{x}) = \boldsymbol{v}^\top \sigma(\boldsymbol{U}\boldsymbol{x})$ where $\sigma(\cdot)$ is the ReLU activation function. $\boldsymbol{v} \in \mathbb{R}^h$ and $\boldsymbol{U} \in \mathbb{R}^{h \times d}$ are parameters[3] and $\boldsymbol{x} \in \mathbb{R}^d$ is the input which is taken from $\mathbb{X} = \{\boldsymbol{x} \in \mathbb{R}^d | \|\boldsymbol{x}\|_2 < b\}$ uniformly. Consider two different networks parameterized with $\{\boldsymbol{U}, \boldsymbol{v}\}$ and $\{\boldsymbol{U}', \boldsymbol{v}'\}$ respectively, and for arbitrarily chosen masks $M_{\boldsymbol{v}} \in \{0, 1\}^h$ and $M_{\boldsymbol{U}} \in \{0, 1\}^{h \times d}$, each element of $\boldsymbol{U}$ and $\boldsymbol{U}'$, $\boldsymbol{v}$ and $\boldsymbol{v}'$ is i.i.d. sampled from a sub-Gaussian distribution sub-$G(0, \sigma_{\boldsymbol{U}}^2)$ and sub-$G(0, \sigma_{\boldsymbol{v}}^2)$ respectively with setting $v_i = v_i'$ when $M_{\boldsymbol{v}, i} = 0$ and $U_{i,j} = U_{i,j}'$ when $M_{\boldsymbol{U}, ij} = 0$. We consider the linear mode connectivity of the two networks and define the difference function between interpolated network and original networks as $z_{\boldsymbol{x}}(\alpha) = (\alpha\boldsymbol{v} + (1 - \alpha)\boldsymbol{v}')^\top \sigma((\alpha\boldsymbol{U} + (1 - \alpha)\boldsymbol{U}')\boldsymbol{x}) - \alpha\boldsymbol{v}^\top \sigma(\boldsymbol{U}\boldsymbol{x}) - (1 - \alpha)\boldsymbol{v}'^\top \sigma(\boldsymbol{U}'\boldsymbol{x})$, $\alpha \in [0, 1]$. The function over all inputs is defined as $z(\alpha) = \frac{1}{|\mathbb{X}|} \int_{\mathbb{X}} z_{\boldsymbol{x}}(\alpha) d\boldsymbol{x}$. We use $|z(\alpha)|$, $\left|\frac{dz(\alpha)}{d\alpha}\right|$ and $\left|\frac{d^2 z(\alpha)}{d\alpha^2}\right|$ to depict the linear mode connectivity, showing the output changes along the $\alpha$ path. With probability $1 - \delta$, it has,*

$$|z(\alpha)| \leq \sqrt{2}b\sigma_{\boldsymbol{v}}\sigma_{\boldsymbol{U}} \log(8h/\delta)\sqrt{h}\sqrt{1 - \rho_{\boldsymbol{U}}}, \tag{5}$$

$$\left|\frac{dz(\alpha)}{d\alpha}\right| \leq 4\sqrt{2}b\sigma_{\boldsymbol{v}}\sigma_{\boldsymbol{U}} \log(24h/\delta)\sqrt{h}(\sqrt{1 - \rho_{\boldsymbol{v}}} + \sqrt{1 - \rho_{\boldsymbol{U}}}), \tag{6}$$

$$\left|\frac{d^2 z(\alpha)}{d\alpha^2}\right| \leq 8b\sigma_{\boldsymbol{v}}\sigma_{\boldsymbol{U}} \log(4h/\delta)\sqrt{h}\sqrt{(1 - \max\{\rho_{\boldsymbol{U}}, \rho_{\boldsymbol{v}}\})}, \tag{7}$$

---

[3]For simplicity and without loss of generality, we omit the bias terms.

*where $\rho_v$ and $\rho_U$ refer to the mask ratios (the proportion of zeros in the mask) of masks $M_v$ and $M_U$ respectively.*

**Remark 3.3** $|z(\alpha)|$ *is the barrier given $\alpha$.* $\left|\frac{dz(\alpha)}{d\alpha}\right|$ *demonstrates the barrier function changes along the interpolation path $\alpha \in [0, 1]$, and the smaller value means smaller changes. If $\left|\frac{dz(\alpha)}{d\alpha}\right| \to 0$, it means that $z(\alpha)$ is a constant, but it does not mean $z(\alpha)$ is a linear function of $\alpha$.* $\left|\frac{d^2 z(\alpha)}{d\alpha^2}\right|$ *reflects the linearity of function $z(\alpha)$, and if $\left|\frac{d^2 z(\alpha)}{d\alpha^2}\right| \to 0$, it means that $z(\alpha)$ is linear w.r.t. $\alpha$.*

Theorem 3.2 preliminarily shows that if the learnable weights (higher $\rho_v$ and $\rho_U$) are reduced by masking some weights from updating (learning in the subspace), LMC can be improved. We will show in the next section how to find such an effective subspace to reduce permutation symmetries and improve LMC.

## 4   TRAINING-TIME NEURON ALIGNMENT BY PARTIALLY FIXING NEURONS

We aim to explore the effective subspaces for improving LMC. The "effective" means that i) learning in this subspace will not hurt the generalization of the trained models; ii) the barrier in LMC can be reduced; iii) the convex fusion of multiple models (i.e., multi-model LMC) can be also improved. In our preliminary attempts, we have studied some existing subspace learning methods such as LoRA (Hu et al., 2021) (a.k.a. learning in intrinsic dimension (Li et al., 2018a)) and model pruning (Liu et al., 2018) and found they can generally improve LMC but not effective enough (results of pruning are in Figure 2 and discussion of LoRA is in subsection C.1). Thus, in this section, we present Training-time Neuron Alignment with Partially Fixed Neurons (TNA-PFN) which is found to be more effective for improving LMC and model fusion.

### 4.1   METHOD FORMULATION

The number of permutation symmetries is numerous since the positions of neurons are not *fixed* and the network is *symmetric*. Hence, we propose to fix some neurons' weights, which will break the network symmetry so the permutations are reduced, and due to the redundancy of neural networks (Liu et al., 2018; Frankle & Carbin, 2018), the accuracy will not be hurt in most cases. An intuitive demonstration is in Figure 1, by fixing some weights of neurons, the number of potential permutations decreases.

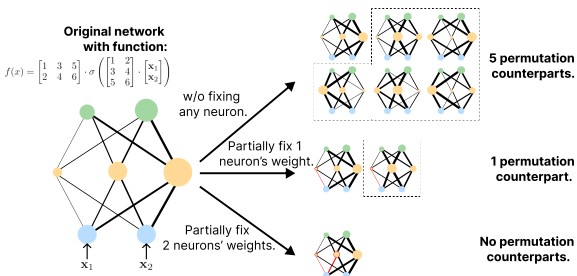

Figure 1: **A simple demonstration showcasing how TNA-PFN can reduce the permutation symmetries.**

Specifically, given an initial network parameterized by a weight vector $\mathbf{w}^0 \in \mathbb{R}^d$. For $\mathbf{w}^0$, we randomly generate a mask for each layer according to the mask ratio $\rho$ (refers to the proportion of zeros in the mask $\mathbf{m}^0$), and the whole mask is $\mathbf{m}^0 \in \{0, 1\}^d$. In $\mathbf{m}^0$, 0 for fixed and 1 for updated, indicating the parameter update status. We individually train $n$ models with different batch orders or datasets. We set each model's initialization as $\mathbf{w}_i \leftarrow \mathbf{w}^0, i \in [n]$. Each model $\mathbf{w}_i, i \in [n]$ conducts the following updates in every SGD iteration:

$$\mathbf{w}_i \leftarrow \mathbf{w}_i - \eta(\mathbf{m}^0 \odot \mathbf{g}_i(\mathbf{w}_i)), \qquad (8)$$

where $\odot$ denotes the element-wise Hadamard product, $\eta$ refers to the learning rate, and $\mathbf{g}_i$ is its gradients of the optimizer, such as SGD or Adam. After training for $E$ epochs, we validate the LMC with respect to the loss or accuracy barriers in Definitions 2.1 and 2.2. The method is notated as TNA-PFN. **Discussion on gradient/model masks.** Applying gradient masks is discovered in previous gradient compression literature of distribution optimization, but our method is different from the previous works in the aspects as follows. ***Motivation difference:*** Gradient compression is proposed for communication efficiency of distributed optimization while we study the training-time neuron alignment problem in LMC. ***Implementation difference:*** Gradient compression uses *different* random top-k gradient masks at each worker and *changes the mask per communication iteration* (Lin et al., 2018; Vogels

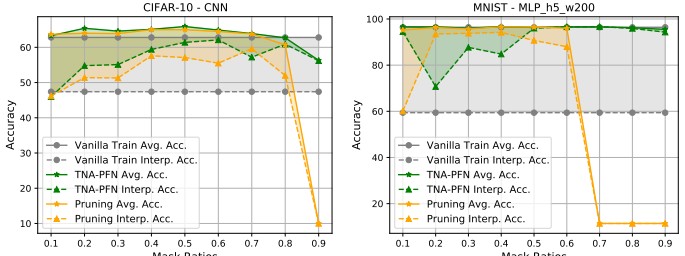

Figure 3: **Left two: Accuracy barriers of MLP under different hidden layers ($h$) and widths ($w$). Right two: Loss landscapes of MLP.** For MLPs, if the barriers exist, TNA-PFN can reduce them. The shadow areas refer to the standard deviations.

et al., 2019); whereas, TNA-PFN uses *unified* random gradient masks at each model, *fixes the mask*, and independently trains the models *without any communications*; and FedPFN/FedPNU (presented in section 5) uses *unified* masks at each client's local training and changes the mask *per global communication round*. **Effect difference:** Since the masks of workers are different and changing, previous gradient compression methods cannot learn in a unified subspace of parameters, while we learn in a subspace by unified gradient masks so that some neuron weights are not updated. We will also discuss the relation between TNA-PFN and model mask (i.e., model pruning) in the following subsection.

## 4.2 Understanding TNA-PFN: Model Pruning and Mask Ratios

According to our "subspace hypothesis", model pruning also has the potential to improve LMC, which is not explored in previous literature. We implement experiments to compare the performances of pruning and TNA-PFN under different mask ratios in Figure 2. Here, we use random pruning at initialization according to the pruning mask ratio $\rho$. *In-*

Figure 2: **Pruning and TNA-PFN under different mask ratios.** The shadow areas mean the accuracy barriers.

*terestingly, it is revealed that pruning can actually facilitate the LMC under mild $\rho$, which further supports our hypothesis.* But when the ratio $\rho$ is high (i.e., 0.8 and 0.9), pruning will result in an untrainable network with nearly zero generalization, while our TNA-PFN also keeps steady performances.

Actually, pruning is a special case of TNA-PFN where the fixed weights are set the same as zero. However, generally in TNA-PFN, the fixed weights' values are different and not zeros, causing different learning dynamics with pruning. An intuitive illustration is in Figure 1 that the fixed values should be different in each neuron to avoid the potential permutation symmetries, but if the fixed weights are pruned, some permutations still exist. In Figure 1, if the red weights in the bottom small network are pruned instead of being fixed, the left two yellow neurons can also be permuted. Another major difference is that high pruning ratios will cause more deaths of neurons, which is detrimental to model learnability, whereas TNA-PFN will keep the neurons activated even if the subspace dimension is low. Additionally, TNA-PFN can be easily incorporated in applications like federated learning (shown in section 5) for improving the global model's generalization while pruning cannot.

Regarding LMC, generally, for both pruning and TNA-PFN, when the mask ratio is higher, the accuracy barriers diminish along with the decrease in the averaged accuracies of independently trained models, showing the *connectivity-accuracy tradeoff*. However, when $\rho$ is set appropriately (e.g., 0.4-0.6 for the CIFAR-10 and CNN setting), both the averaged accuracy and LMC can be improved for pruning and TNA-PFN. The accuracy improvement of random pruning at initialization also validates the observations in Liu et al. (2018), and in this paper, we make the contribution by extending the power of pruning to the improvement of linear mode connectivity.

## 4.3 Experiments on Linear Mode Connecitivity

In this subsection, we will conduct experiments to validate the effectiveness of TNA-PFN in improving LMC. If not mentioned otherwise, the mask ratio $\rho$ of TNA-PFN is 0.4 (the hyperparameter which is mild across various settings).

**Different model depths, widths, and architectures.** In Figure 3, we conduct experiments on MLP with different hidden layers and widths. For MNIST (LeCun & Cortes, 2010), we find shallower and

Table 1: **The performances of post-matching methods after TNA-PFN.** Interpolated Accuracy (Interp. Acc.) means the accuracy of the linearly interpolated model, i.e., $\mathcal{A}(0.5\mathbf{w}_1 + 0.5\mathbf{w}_2)$. "Iter." refers to the number of iterations in the post-matching methods, reflecting the computation costs.

| | CIFAR-10 | | | | MNIST | | | |
| | MLP_h2_w200 | | ResNet20 | | MLP_h5_w200 | | MLP_h6_w200 | |
| Metrics\Methods | TNA-PFN | Vanilla train | TNA-PFN | Vanilla train | TNA-PFN | Vanilla train | TNA-PFN | Vanilla train |
|---|---|---|---|---|---|---|---|---|
| Interp. Acc. w/o Post-matching | 43.7±0.4 | 31.9±2.4 | 46.2±4.7 | 36.1±4.3 | 84.8±8.2 | 59.4±24.2 | 87.5±8.9 | 63.7±15.6 |
| Interp. Acc. after 10 Iter. of SA | 43.7±0.4 | 32.2±2.2 | 46.2±4.7 | 36.7±3.4 | 85.4±8.0 | 59.7±24.2 | 87.7±9.1 | 64.9±14.4 |
| Interp. Acc. after 100 Iter. of SA | 43.7±0.4 | 31.9±2.4 | 46.2±4.7 | 36.1±4.3 | 86.9±7.6 | 60±24.1 | 88.2±7.9 | 64.2±15.1 |
| Interp. Acc. after WM | 48.5±0.9 | 44.7±1.3 | 53.6±2.5 | 53.7±2.9 | 97.1±0.2 | 96.9±0.3 | 96.9±0.4 | 96.8±0.3 |
| Required Iter. in WM | 4.8±1.5 | 5.2±1.0 | 2.5±0.2 | 4.6±0.5 | 7.6±3.8 | 10.4±1.2 | 7.33±4.2 | 11.2±1.8 |

wider networks will not cause barriers, which is consistent with the previous observations (Entezari et al., 2022). For CIFAR-10 (Krizhevsky et al., 2009), the barriers always exist under various depths and widths. Our proposed TNA-PFN can obviously reduce the accuracy barriers from 0.3-0.4 to 0.1, and we also visualize the loss landscapes, which illustrate the barrier reductions. To see more results and illustrations, please refer to subsection C.2.

We study the LMC of simple CNN and ResNets and present the results in Figure 4. ResNets (He et al., 2016) have higher barriers than simple CNN, and the barriers are exacerbated when the networks are deeper or wider. It is suggested that TNA-PFN can lower the barriers under different architectures.

Generally, we observe that TNA-PFN has more dominant advantages when the models are wider, and the observations are: (1) the second figure in Figure 3: for CIFAR-10 when width increases, the barriers of vanilla training go up while the barriers of TNA-PFN go down; (2) Figure 4: the barrier reduction of TNA-PFN is more obvious for WRN56 compared with ResNet56. The reason is TNA-PFN can reduce the potential of permutation invariance within each layer by fixing some weights, but it will have loose regularization on the layer-to-layer relationship, so if the network goes deeper, its advantage will decrease. Considering the layer-to-layer effects on training-time neuron alignment can be an interesting future work.

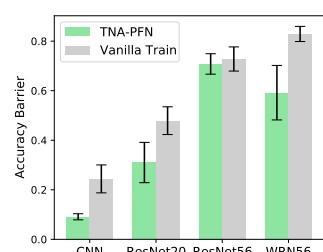

Figure 4: **Accuracy barriers under different model architecture.** WRN56 abbreviates for WideResNet56. CIFAR-10.

**The role of post-hoc neuron alignment methods after training-time alignment.** We consider simulated annealing (SA) (Entezari et al., 2022) and weight matching (WM) (Ainsworth et al., 2022) after TNA-PFN in Table 1. SA requires large computations, and we notice the improvements are also marginal. Under limited computation budgets (10 or 100 iterations), we find that TNA-PFN can reach a higher result than vanilla training after SA. For WM, it is indicated that after TNA-PFN, the required iterations are shortened while the interpolated accuracies are similar. The results reveal that training-time neuron alignment can reduce the costs of post-matching and remain similar or even better post-matched LMC.

**More deep learning tasks.** We conduct more experiments beyond vision tasks and display the results in Table 2. *Polynomial approximation task* (Peña et al., 2023; von Oswald et al., 2019): we use an MLP with one hidden layer to approximate the second and third polynomial functions:

Table 2: **Results of loss barriers on more deep learning tasks.**

| Methods\Datasets | 2nd Polynomial | 3rd Polynomial | IMDb |
|---|---|---|---|
| Vanilla Train | 0.268±0.061 | 0.0554±0.047 | 0.710±0.17 |
| TNA-PFN | 0.0381±0.0096 | 0.0355±0.023 | 0.375±0.17 |

$y = 2x^2 - 1, y = (x - 3)^3$. *Sentiment analysis of text* (Liu et al., 2022b): we use an LSTM (Graves & Graves, 2012) to predict the sentiment of IMDb reviews (Maas et al., 2011). It can be seen that the loss barriers are decreased by training-time alignment under both polynomial approximation and sentiment analysis tasks. We also implement the experiments on a large-scale dataset, the subset of ImageNet (Deng et al., 2009; tin, Accessed: 2023). The result is shown in Table 9 of Appendix.

**Results under multi-model fusion.** We study the LMC of multi-model fusion and the results are shown in Table 3. We consider the connectivity of 5 independently trained models by assigning a uniformly weighted fusion after training. We test the generalization of the fused model as interpolated accuracy and compared it with the averaged accuracy of independent models. It is evident that after the training-time alignment, the interpolated accuracies are largely promoted by up to 152% and the barriers are much lower with a maximal reduction of 84.9%, showing TNA-PFN's prospects in applications like federated learning. It is also intriguing to observe that the averaged accuracies also increase after TNA-PFN. We explain this phenomenon that partially fixing some weights may play

Table 3: **Linear mode connectivity of multi-model fusion.** The number of models is 5.

| Datasets / Models | Metrics | Vanilla Train | TNA-PFN |
|---|---|---|---|
| CIFAR-10 / CNN | Avg. Acc. | $63.1 \pm 0.6$ | $65.5 \pm 0.3$ |
| | Interp. Acc. | $21.3 \pm 9.1$ | $48.3 \pm 7.2$ |
| | Acc. Barrier | $0.663 \pm 0.14$ | $0.264 \pm 0.11$ |
| CIFAR-10 / MLP_h2_w200 | Avg. Acc. | $44.2 \pm 0.5$ | $48.4 \pm 0.5$ |
| | Interp. Acc. | $21.6 \pm 1.9$ | $36.6 \pm 1.4$ |
| | Acc. Barrier | $0.511 \pm 0.043$ | $0.245 \pm 0.035$ |
| MNIST / MLP_h5_w200 | Avg. Acc. | $96.5 \pm 0.3$ | $96.5 \pm 0.2$ |
| | Interp. Acc | $34.5 \pm 15.5$ | $87.1 \pm 9.4$ |
| | Acc. Barrier | $0.643 \pm 0.16$ | $0.0974 \pm 0.096$ |

Table 4: **Linear mode connectivity of non-random initialization.** The initialized model is first trained on a disjoint dataset for 0.5 epoch. The dataset is CIFAR-10.

| Models | Metrics | TNA-PFN | Vanilla Train |
|---|---|---|---|
| CNN | Avg. Acc. | $65.8 \pm 0.2$ | $64.4 \pm 0.7$ |
| | Interp. Acc. | $63.3 \pm 1.2$ | $52.8 \pm 2.3$ |
| | Acc. Barrier | $0.0413 \pm 0.018$ | $0.181 \pm 0.027$ |
| | Loss Barrier | $0.0762 \pm 0.038$ | $0.306 \pm 0.058$ |
| ResNet20 | Avg. Acc. | $67.1 \pm 2.4$ | $69.4 \pm 0.8$ |
| | Interp. Acc. | $48.5 \pm 2.0$ | $42.2 \pm 8.5$ |
| | Acc. Barrier | $0.277 \pm 0.019$ | $0.393 \pm 0.12$ |
| | Loss Barrier | $0.453 \pm 0.044$ | $0.675 \pm 0.21$ |

the role of regularization and for some models with redundant neurons, this regularization can also help in generalization.

**Results under non-random initializations.** We also examine whether training-time alignment can help when the initializations are not random, which commonly occurs in the pretraining-finetuning paradigm and federated learning. We first trained a model from random initialization on a dataset that shares the same distribution but is disjoint with the trainset for 0.5 epoch. Then, the trained model is set as the initialization. As presented in Table 4, TNA-PFN is also beneficial to improve LMC under non-random initialization, and it nearly clears the barriers under the setting of CNN and CIFAR-10. However, we notice a slight decline in average accuracy when the model is ResNet20.

**Layer-wise analysis.** We conduct a layer-wise analysis of TNA-PFN to see which layer matters most in improving LMC in Figure 7 of Appendix.

## 5 EXTENDING TRAINING-TIME ALIGNMENT IN FEDERATED LEARNING

Federated learning requires model fusion on the server, and it meets neuron alignment problems during training (Wang et al., 2020b; Li et al., 2022; Yurochkin et al., 2019). Previous methods utilize post-hoc matching methods on the server, and they usually require large computations. In this paper, we extend training-time neuron alignment methods in federated learning (FL) to improve the global model's generalization to showcase its potential in applications. Due to the space limit, the preliminary of FL is in Appendix D.

### 5.1 METHODS

In this subsection, we propose two variants of TNA-PFN in FL, the first is called Federated Learning with Partially Fixed Neurons (FedPFN) and the second is Federated Learning with Progressive Neuron Updating (FedPNU).

**FedPFN** (pseudo-code in Algorithm 1). There are $T$ communication rounds in FL. During FL training, in communication round $t \in [T]$, the central server generates a random mask $\mathbf{m}^t \in \{0, 1\}^d$ according to the masking ratio $\rho$. Also, the central server generates the global model $\mathbf{w}^t$ by the global aggregation scheme (e.g., FedAvg (McMahan et al., 2017)) and sends $\mathbf{m}^t$ and $\mathbf{w}^t$ to the clients.
Clients initialize their local models as the received global model, $\mathbf{w}_i^t \leftarrow \mathbf{w}^t$. Client $i$ conducts SGD updates with mask $\mathbf{m}^t$, so that the masked neuron weights are fixed at this round. The SGD updates are as follows for $E$ epochs,

$$\mathbf{w}_i^t \leftarrow \mathbf{w}_i^t - \eta_l(\mathbf{m}^t \odot \mathbf{g}_i(\mathbf{w}_i^t)), \tag{9}$$

where $\odot$ denotes the element-wise (Hadamard) product and $\eta_l$ refers to the local learning rate. By applying FedPFN, during local training, all clients learn in the same effective subspace so model drifts and permutation invariance issues can be relieved. Besides, the neuron mask $\mathbf{m}^t$ changes from round to round, so all the neurons can be evenly trained, and it will break the connectivity-accuracy tradeoff observed in LMC.

**FedPNU** (pseudo-code in Algorithm 2). In FedPNU, we additionally consider a reversed mask $\hat{\mathbf{m}}^t$ of $\mathbf{m}^t$. During training, the clients first train with mask $\mathbf{m}^t$ according to Equation 9 for the half local training, i.e., $\text{int}(\frac{E}{2})$ epochs; then, they train with the reversed mask $\hat{\mathbf{m}}^t$ for the remaining $E - \text{int}(\frac{E}{2})$ epochs. In FedPNU, the clients progressively train the networks in a subspace and the accordingly complementary subspace, by which the neurons of local models are more aligned.

For more discussion about the related works of subspace and partial training in federated learning, please refer to Appendix E.

We note that FedPFN and FedPNU are lightweight and flexible since they only add a gradient mask before the optimizer's updates, so they are orthogonal to current FL algorithms (especially server-side global model fusion schemes). We will show they can be incorporated into existing FL methods for further improving performances.

Table 5: **Top-1 test accuracy (%) achieved by comparing the FL methods on three datasets with different model architectures** ($E = 3$). **Bold** fonts highlight the best two methods in each setting.

| Datasets | FashionMNIST | | | | CIFAR-10 | | | | CIFAR-100 | | | |
|---|---|---|---|---|---|---|---|---|---|---|---|---|
| dir | 100 | | 0.1 | | 100 | | 0.1 | | 100 | | 0.1 | |
| Methods\Models | MLP | LeNet | MLP | LeNet | CNN | ResNet | CNN | ResNet | CNN | ResNet | CNN | ResNet |
| FedAvg | 88.7±0.5 | 90.5±0.1 | 81.7±2.7 | 83.5±3.8 | 65.4±1.2 | 73.4±2.1 | 57.5±1.3 | 50.9±1.8 | 18.9±0.9 | 26.4±0.4 | 22.6±0.9 | 28.5±1.3 |
| FedPFN | 88.8±0.1 | **90.6±0.1** | 81.8±1.7 | 84.9±2.8 | 66.9±0.6 | 73.7±1.3 | **62.2±0.5** | 51.4±0.7 | 20.9±0.7 | 27.3±0.3 | 24.9±1.2 | 34.5±2.4 |
| FedPNU | 88.7±0.2 | 90.4±0.2 | **83.2±0.8** | **86.6±1.0** | **67.5±0.3** | 73.5±2.5 | **61.3±0.4** | **55.9±1.6** | 22.1±0.5 | 29.4±0.0 | 24.8±0.2 | **35.3±1.5** |
| FedProx | 88.0±0.1 | 90.0±0.2 | 82.6±0.9 | 85.8±0.7 | 65.4±0.9 | 65.5±0.8 | 59.7±1.1 | 49.9±2.1 | **27.7±0.5** | 26.7±0.4 | 24.7±0.0 | 23.0±1.5 |
| FedProx+FedPFN | 86.9±0.1 | 89.5±0.1 | 81.4±1.3 | 85.0±0.8 | 66.9±1.0 | 60.3±0.7 | 60.3±0.5 | 51.8±0.5 | **27.7±0.7** | 19.6±0.1 | 24.4±0.0 | 17.3±0.6 |
| FedProx+FedPNU | 86.2±0.1 | 89.2±0.1 | 81.0±1.3 | 84.6±0.6 | 67.2±0.1 | 57.2±1.0 | 60.0±0.3 | 49.7±0.3 | 24.7±0.2 | 15.9±0.3 | 22.9±1.0 | 16.6±0.9 |
| FedDF | **89.1±0.1** | 90.3±0.2 | 81.3±2.8 | 86.0±1.9 | 66.3±0.8 | **75.6±3.3** | 57.6±3.0 | 55.2±1.4 | 21.4±0.3 | 28.5±1.0 | 24.2±0.2 | 31.2±1.2 |
| FedDF+FedPFN | **88.9±0.2** | 90.5±0.1 | 80.7±3.3 | **86.4±2.0** | **67.9±0.4** | 73.0±1.2 | 59.3±3.6 | 54.4±5.1 | **27.8±0.8** | **31.0±1.3** | **27.0±0.2** | 31.1±1.2 |
| FedDF+FedPNU | 88.7±0.1 | **90.7±0.2** | 82.1±2.4 | **86.4±1.8** | 66.4±0.8 | **74.1±1.3** | 60.0±2.4 | **57.1±4.1** | 22.2±0.9 | **30.8±1.2** | **25.8±0.7** | 35.2±1.2 |

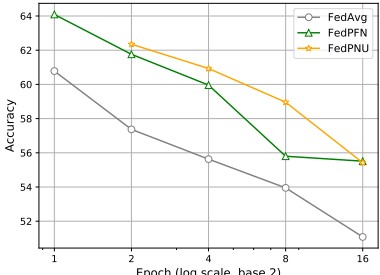

Figure 5: **Performances of FedPFN and FedPNU under different local epochs.** CIFAR-10 with dir = 0.1 and the model is CNN.

Figure 6: **Performances of FedPFN and FedPNU under different mask ratios.** CIFAR-10, CNN, and $E = 3$.

## 5.2 EXPERIMENTS

**Settings and baselines.** We use three datasets to verify the algorithms: FashionMNIST (Xiao et al., 2017), CIFAR-10, and CIFAR-100 (Krizhevsky et al., 2009). We adopt the Dirichlet sampling to generate non-IID data (a.k.a., heterogeneous data) for each client, which is widely used in FL literature (Lin et al., 2020; Chen & Chao, 2021). The Dirichlet sampling considers a class-imbalanced data heterogeneity, controlled by the hyperparameter "dir", the smaller, the more heterogeneous. We vary "dir" in range $[100, 0.5, 0.1]$, which respectively means IID, moderately non-IID, and extremely non-IID data. For FashionMNIST, the models are MLP_h2_w200 and LeNet5 (LeCun et al., 1998); for CIFAR-10 and CIFAR-100, the models are simple CNN (Li et al., 2023) and ResNet20 (He et al., 2016). For client-side methods, we consider vanilla training and FedProx (Li et al., 2020a); for the server-side algorithms, we consider FedAvg (McMahan et al., 2017) and FedDF (Lin et al., 2020). If not mentioned otherwise, the number of clients in the experiments is 20 and full client selection is applied. For more implementation details, please refer to Appendix A.

**Different datasets and models.** In Table 5, we demonstrate the results under different datasets, data heterogeneity, and models. We note that the vanilla FedPFN/FedPNU are actually FedAvg+FedPFN/FedPNU, and we also combine our methods with server-side approach FedDF and client-side FedProx. FedPFN and FedPNU consistently improve over FedAvg, showing that incorporating the training-time alignment method can boost model fusion in FL. It can be seen that the variants of our methods usually achieve the best results. Specifically, they can strengthen FedDF to reach a higher performance. However, we find our methods are not always compatible with FedProx, especially when FedProx is worse than FedAvg; but in some cases when FedProx works well, our methods can also strengthen it (e.g., CIFAR-10 with CNN).

**Different number of clients.** We scale the number of clients in the range $[30, 60, 90]$ and apply partial client selections when the number is 90. From Table 6, it is found that our methods can still improve the global model's generalization when scaling up the clients, which showcases the effectiveness of training-time neuron alignment methods in improving multi-model linear mode connectivity.

Table 6: **Results about different numbers of clients and partial selections.** CIFAR-10 with dir = 0.5 and $E = 3$, the model is CNN.

| Methods | Number of clients (selection ratio) | | | | |
|---|---|---|---|---|---|
| | 30 (1.0) | 60 (1.0) | 90 (0.4) | 90 (0.6) | 90 (1.0) |
| FedAvg | 63.6±1.2 | 62.5±0.7 | 60.8±0.4 | 61.4±0.7 | 61.6±0.5 |
| FedPFN | 65.6±0.1 | 64.7±0.3 | 62.9±0.4 | 63.6±0.5 | 64.0±0.4 |
| FedPNU | 65.2±0.2 | 63.2±0.7 | 62.0±0.9 | 62.3±0.3 | 62.3±0.7 |

**Different local epochs.** We verify the TNA variants in FL under different local epochs in Figure 5. We find that the improvements are also strong when there are more local updates. It is observed that FedPNU is more robust regarding local epochs, and this is because it learns in the complementary

subspaces progressively, reducing the negative effects of subspaces on accuracy. Similar reasons are also for why FedPNU is robust when the mask ratio is as high as 0.9 in Figure 6.

**The effects of mask ratios for FedPFN and FedPNU.** From Figure 6, it is shown that FedPFN benefits under smaller subspaces (higher mask ratios) but falls short when the subspace is too small (ratio $\rho = 0.9$); whereas FedPNU is robust across all mask ratios due to its progressive learning.

**Comparison with pruning and fixed masks.** We make an ablation study on the design of FedPFN. We compare FedPFN with the TNA-PFN variant denoted as FedPFN (fixed) in which we fix the neuron mask $\mathbf{m}^t$ in every round ($\mathbf{m}^t = \mathbf{m}^{t-1} = \mathbf{m}^0$). We also implement the setting where the random pruning is applied at

Table 7: **Results of random initialization pruning in FL and fixing FedPFN's mask.** CIFAR-10 with dir = 0.3 and $E = 3$.

| Models\Methods | FedAvg | FedPFN | FedPFN (fixed) | FedPruning |
|---|---|---|---|---|
| CNN | 64.8 ± 1.0 | 65.7 ± 1.0 | 64.9 ± 1.0 | 63.7 ± 1.1 |
| ResNet20 | 72.0 ± 0.7 | 72.4 ± 0.5 | 71.3 ± 1.3 | 70.2 ± 1.3 |

initialization before FL training, named as FedPruning. Table 7 presents the results. Although we find pruning can improve LMC in subsection 4.2, it will cause generalization degradation in FL due to the connectivity-accuracy tradeoff. Also, if we incorporate TNA-PFN by keeping the same neuron mask during FL training, it will have marginal or even negative improvements. The above findings indicate that FL is sensitive in the subspaces and further validate the rationale of our devised methods. We include more results and illustrations in subsection C.3.

## 6 RELATED WORKS

**Linear Mode Connectivity.** Linear mode connectivity refers to the phenomenon that there exists a loss (energy) barrier along the linear interpolation path of two networks, in the cases where i) the two networks have the same initialization and are trained on the same dataset but with different random seeds (data shuffles and augmentations) (Ainsworth et al., 2022); ii) the two networks are with different initializations but are trained on the same dataset (Entezari et al., 2022); iii) the two networks are the initial network and the final trained network (Vlaar & Frankle, 2022). Specifically, Adilova et al. (2023) examines the linear mode connectivity of different layers. Vlaar & Frankle (2022) studies the relationship between generalization and the initial-to-final linear mode connectivity. Frankle et al. (2020) connects linear mode connectivity with the lottery ticket hypothesis and finds better connectivity can result in better pruning performances. Zhao et al. (2020) bridges mode connectivity and adversarial robustness. Some works try to extend mode connectivity beyond "linear", e.g., searching for a non-linear low-loss path (Draxler et al., 2018) or studying mode connectivity under spurious attributes (Lubana et al., 2023).

**Permutation Invariance and Model Fusion.** Permutation invariance (a.k.a. permutation symmetry) refers to the property of neural networks that the positions of neurons can be permuted without changing its function (Simsek et al., 2021; Hecht-Nielsen, 1990), and it is believed to be the primary cause of loss barrier in linear mode connectivity (Entezari et al., 2022; Ainsworth et al., 2022). Entezari et al. (2022) hypothesizes that if taking the permutation invariance into consideration, all solutions can be mapped into the same low-loss basin with connectivity. Further, Ainsworth et al. (2022) validates this hypothesis by using "re-basin" which aims to find the appropriate permutation matrices to map the networks into the same basin. Other methods are also utilized to match the neurons for better model fusion, such as optimal transport (Singh & Jaggi, 2020), Bayesian nonparametric technique (Yurochkin et al., 2019; Wang et al., 2020a), Hungarian algorithm (Tatro et al., 2020), graph matching (Liu et al., 2022a), and implicit Sinkhorn differentiation (Peña et al., 2023). We note that all these methods are for post-matching after training, while we focus on training-time neuron alignment. Note that the previous work of PAN (Li et al., 2022) shares a similar motivation with ours. PAN uses position-aware encoding for intermediate activations which is orthogonal to our methods, and it only focuses on FL, while we study both LMC and model fusion in FL.

In addition, we include more discussion of related works in Appendix E.

## 7 CONCLUSION

In this paper, we revisit neuron alignment and linear mode connectivity from the training-time perspective. We propose the hypothesis that if the networks can be learned in an effective subspace, the linear mode connectivity can be improved. We verify the hypothesis theoretically and empirically and find random pruning at initialization can actually improve connectivity. We further propose the training-time neuron alignment method which randomly fixes the neuron weights during training to reduce the potential of permutation symmetries. We then devise two variants of the training-time alignment method in federated learning for improving the global model's generalization.

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

# Appendix

In this appendix, we provide the details omitted in the main paper and more analyses and discussions.

- Appendix A: details of experimental setups (cf. section 4 and section 5 of the main paper).
- Appendix B: detailed proof of Theorem 3.2 (cf. section 3 of the main paper).
- Appendix C: additional results and analyses (cf. section 4 and section 5 of the main paper).
- Appendix D: preliminary of federated learning (cf. section 2 and section 5).
- Appendix E: more discussions about the related works (cf. section 6 of the main paper).

## A    IMPLEMENTATION DETAILS

Epochs lr in LMC datasets: poly
models: MLP CNN ResNet (without BN)
In this section, we provide the additional implementation details in section 4 and section 5.

### A.1    DATASETS

MNIST. CIFAR-10. Poly. IMDb. FashionMNIST. CIFAR-100.
**MNIST** (LeCun & Cortes, 2010) comprises a collection of 70,000 handwritten digits (0-9), divided into 60,000 training images and 10,000 testing images. Each image is grayscale and has a dimension of 28x28 pixels. **CIFAR-10** (Krizhevsky et al., 2009) consists of 60,000 32x32 color images, evenly distributed across 10 different classes or labels, such as airplanes, automobiles, birds, cats, etc., each represented by 6,000 images. The dataset is split into 50,000 training images and 10,000 test images. **The polynomial approximation dataset** (Peña et al., 2023; von Oswald et al., 2019) is the synthetic dataset of the second and third polynomial functions: $y = 2x^2 - 1, y = (x - 3)^3$. The input of the second polynomial function is uniformly generated from $x \in [-1.0, 1.0]$ with 100 data points; and the input of the third polynomial function is uniformly generated from $x \in [2.0, 4.0]$ with 100 data points. Each $y$ label in both the second and the third polynomial datasets is added by a random Gaussian noise with zero mean and 0.05 std. The **IMDb** (Internet Movie Database) (Maas et al., 2011) dataset is a popular dataset used in Natural Language Processing (NLP) and sentiment analysis tasks. It consists of 50,000 movie reviews, evenly split into 25,000 reviews for training and 25,000 reviews for testing, each labeled as either positive or negative. **FashionMNIST** (Xiao et al., 2017) is a dataset designed as a more advanced replacement for the MNIST dataset, suitable for benchmarking machine learning models. It consists of 70,000 images divided into 60,000 training samples and 10,000 test samples. Each image is a 28x28 grayscale representation of fashion items from 10 different classes, including shirts, trousers, sneakers, etc. The **CIFAR-100** dataset (Krizhevsky et al., 2009) is similar to the CIFAR-10 dataset but more challenging as it contains 100 different classes grouped into 20 superclasses. It contains 60,000 32x32 color images, with 600 images per class, divided into 50,000 training images and 10,000 test images. This dataset is primarily used for developing and evaluating more sophisticated image classification models.

### A.2    MODELS

**CNN and MLP.** The simple CNN for CIFAR-10 and CIFAR-100 is a convolution neural network model with ReLU activations which consists of 3 convolutional layers followed by 2 fully connected layers. The first convolutional layer is of size (3, 32, 3) followed by a max pooling layer of size (2, 2). The second and third convolutional layers are of sizes (32, 64, 3) and (64, 64, 3), respectively. The last two connected layers are of sizes (64*4*4, 64) and (64, num_classes), respectively.

The MLP model MLP_h2_w200 stands for an MLP with 2 hidden layers and a width of 200 in each layer. We vary h and w in Figure 3 to see the barriers in linear mode connectivity. We use MLP_h2_w200 for the MLP model in Table 5.

**ResNets.** We followed the model architectures used in (Li et al., 2018b). The number of the model names means the number of layers of the models. Naturally, the larger number indicates a deeper network. For WRN56 in Figure 4, it is an abbreviation of Wide-ResNet56-4, where "4" refers to four times as many filters per layer. The ResNets used in Table 5 are ResNet20 for CIFAR-10 and CIFAR-100. It is notable that since the batch normalization layers will have abnormal effects on model fusion (Li et al., 2020b; Lin et al., 2020), following Adilova et al. (2023), we remove all the batch normalization layers from the ResNets.

### A.3 RANDOMNESS

In all experiments, we implement the experiments three times with different random seeds and report the averaged results with standard deviations.

For the experiments in linear mode connectivity, within a set of experiments, we generate an initial model according to the random seed $a$ and conduct training, then, we set the random seed as $a+1$ and load the initial model from random seed $a$ and conduct another independent training; afterward, the linear connectivity of the two models are tested.

For the experiments in federated learning. Given a random seed, we set torch, numpy, and random functions as the same random seed to make the data partitions and other settings identical. To make sure all algorithms have the same initial model, we save an initial model for each architecture and load the saved initial model at the beginning of one experiment. Also, for the experiments with partial participation, the participating clients in each round are vital in determining the model performance, and to guarantee fairness, we save the sequences of participating clients in each round and load the sequences in all experiments. This will make sure that, given a random seed and participation ratio, every algorithm will have the same sampled clients in each round.

### A.4 EVALUATION

**Linear mode connectivity.** We validate all the accuracy and loss barriers on the test datasets to indicate the model generalization.

**Federated learning.** We evaluate the global model performance on the test dataset of each dataset. The test dataset is mostly class-balanced and can reflect the global learning objective of a federated learning system. Therefore, the performance of the model on the test set can indicate the generalization performance of global models (Li et al., 2023; Lin et al., 2020). In each experiment, we run 100 rounds and take the average test accuracy of the last 5 rounds as the final test accuracy.

### A.5 HYPERPARAMETER

**Linear mode connectivity.** For CIFAR-10 and MNIST, We set a fixed learning rate of 0.1 and use the SGD optimizer with a weight decay of 5e-4 and momentum of 0.9; the number of learning epochs is 10. For the Polynomial datasets, the learning rate is 0.05 for 100 epochs. For the IMDb dataset, the learning rate is 0.0005 for 20 epochs.

**Federated learning.** We set the initial learning rates as 0.08 in CIFAR-10 and FashionMNIST and set it as 0.05 in CIFAR-100. Following (Li et al., 2023; Chen & Chao, 2021), we set a decaying learning rate scheduler in all experiments; that is, in each round, the local learning rate is 0.99*(the learning rate of the last round). We set the weight decay factor as 5e-4. We set SGD optimizer as the clients' local solver and set momentum as 0.9.

For the server-side optimizer FedDF, the server-side learning rate is 0.01 and the number of epochs is 20. We set $\mu = 0.001$ for FedProx.

### A.6 PSEUDO-CODES

We present the pseudo-codes of the federated learning methods FedPFN and FedPNU in Algorithm 1 and Algorithm 2.

---

**Algorithm 1 FedPFN**: Federated Learning with Partially Fixed Neurons

---

**Input**: clients $\{1, \ldots, n\}$, mask ratio $\rho$, comm. round $T$, local epoch $E$, initial global model $\mathbf{w}_g^1$;

**Output**: final global model $\mathbf{w}_g^T$;

1: **for** each round $t = 1, \ldots, T$ **do**
2:     # Client updates
3:     **for** each client $i, i \in [n]$ **in parallel do**
4:         Receive global model $\mathbf{w}_g^t$ and neuron mask $\mathbf{m}^t$;
5:         Set local model $\mathbf{w}_i^t \leftarrow \mathbf{w}_g^t$;
6:         Compute $E$ epochs of client local training by Equation 9:
7:             $\mathbf{w}_i^t \leftarrow \mathbf{w}_i^t - \eta_l(\mathbf{m}^t \odot \mathbf{g}_i(\mathbf{w}_i^t))$;
8:     **end for**
9:     # Server updates
10:    The server samples $m$ clients and receive their models $\{\mathbf{w}_i^t\}_{i=1}^m$;
11:    Obtain the global model by FedAvg:
12:       $\mathbf{w}_g^{t+1} \leftarrow \sum_{i=1}^m \lambda_i \mathbf{w}_i^t$, where $\lambda_i$ is the aggregation weight of client $i$;
13:    Randomly generate the new neuron mask $\mathbf{m}^{t+1}$ according to the ratio $\rho$.
14: **end for**
15: Obtain the final global model $\mathbf{w}_g^T$.

---

**Algorithm 2 FedPNU**: Federated Learning with Progressive Neuron Updating

---

**Input**: clients $\{1, \ldots, n\}$, mask ratio $\rho$, comm. round $T$, local epoch $E$, initial global model $\mathbf{w}_g^1$;

**Output**: final global model $\mathbf{w}_g^T$;

1: **for** each round $t = 1, \ldots, T$ **do**
2:     # Client updates
3:     **for** each client $i, i \in [n]$ **in parallel do**
4:         Receive global model $\mathbf{w}_g^t$ and neuron mask $\mathbf{m}^t$;
5:         Set local model $\mathbf{w}_i^t \leftarrow \mathbf{w}_g^t$ and compute the reverse mask $\hat{\mathbf{m}}^t$ of $\mathbf{m}^t$;
6:         Compute $\text{int}(\frac{E}{2})$ epochs of client local training by Equation 9:
7:             $\mathbf{w}_i^t \leftarrow \mathbf{w}_i^t - \eta_l(\mathbf{m}^t \odot \mathbf{g}_i(\mathbf{w}_i^t))$;
8:         Compute $E$ - $\text{int}(\frac{E}{2})$ epochs of client local training by Equation 9:
9:             $\mathbf{w}_i^t \leftarrow \mathbf{w}_i^t - \eta_l(\hat{\mathbf{m}}^t \odot \mathbf{g}_i(\mathbf{w}_i^t))$;
10:    **end for**
11:    # Server updates
12:    The server samples $m$ clients and receive their models $\{\mathbf{w}_i^t\}_{i=1}^m$;
13:    Obtain the global model by FedAvg:
14:       $\mathbf{w}_g^{t+1} \leftarrow \sum_{i=1}^m \lambda_i \mathbf{w}_i^t$, where $\lambda_i$ is the aggregation weight of client $i$;
15:    Randomly generate the new neuron mask $\mathbf{m}^{t+1}$ according to the ratio $\rho$.
16: **end for**
17: Obtain the final global model $\mathbf{w}_g^T$.

---

# B   PROOF OF THEOREM 3.2

We first recap the Theorem 3.2 for convenience and provide the proof.

**Theorem B.1** *We define a two-layer neural network with ReLU activation, and the function is $f_{\boldsymbol{v},\boldsymbol{U}}(\boldsymbol{x}) = \boldsymbol{v}^\top \sigma(\boldsymbol{U}\boldsymbol{x})$ where $\sigma(\cdot)$ is the ReLU activation function. $\boldsymbol{v} \in \mathbb{R}^h$ and $\boldsymbol{U} \in \mathbb{R}^{h \times d}$ are parameters[4] and $\boldsymbol{x} \in \mathbb{R}^d$ is the input which is taken from $\mathbb{X} = \{\boldsymbol{x} \in \mathbb{R}^d | \|\boldsymbol{x}\|_2 < b\}$ uniformly. Consider two different networks parameterized with $\{\boldsymbol{U}, \boldsymbol{v}\}$ and $\{\boldsymbol{U}', \boldsymbol{v}'\}$ respectively, and for arbitrarily chosen masks $\boldsymbol{M}_{\boldsymbol{v}} \in \{0,1\}^h$ and $\boldsymbol{M}_{\boldsymbol{U}} \in \{0,1\}^{h \times d}$, each element of $\boldsymbol{U}$ and $\boldsymbol{U}'$, $\boldsymbol{v}$ and $\boldsymbol{v}'$ is i.i.d. sampled from a sub-Gaussian distribution sub-$G(0, \sigma_{\boldsymbol{U}}^2)$ and sub-$G(0, \sigma_{\boldsymbol{v}}^2)$ respectively with setting $v_i = v_i'$ when $M_{\boldsymbol{v},i} = 0$ and $U_{i,j} = U_{i,j}'$ when $M_{\boldsymbol{U},ij} = 0$. We consider the linear mode connectivity of the two networks and define the difference function between interpolated network and original networks as $z_{\boldsymbol{x}}(\alpha) = (\alpha\boldsymbol{v} + (1-\alpha)\boldsymbol{v}')^\top \sigma((\alpha\boldsymbol{U} + (1-\alpha)\boldsymbol{U}')\boldsymbol{x}) - \alpha\boldsymbol{v}^\top\sigma(\boldsymbol{U}\boldsymbol{x}) - (1-$*

---

[4]For simplicity and without loss of generality, we omit the bias terms.

$\alpha)\boldsymbol{v'}^{\top}\sigma(\boldsymbol{U'}\boldsymbol{x})$, $\alpha \in [0,1]$. *The function over all inputs is defined as* $z(\alpha) = \frac{1}{|\mathbb{X}|}\int_{\mathbb{X}} z_{\boldsymbol{x}}(\alpha)d\boldsymbol{x}$. *We use* $|z(\alpha)|$, $\left|\frac{dz(\alpha)}{d\alpha}\right|$ *and* $\left|\frac{d^2z(\alpha)}{d\alpha^2}\right|$ *to depict the linear mode connectivity, showing the output changes along the* $\alpha$ *path. With probability* $1 - \delta$, *it has,*

$$|z(\alpha)| \leq \sqrt{2}b\sigma_{\boldsymbol{v}}\sigma_{\boldsymbol{U}}\log(8h/\delta)\sqrt{h}\sqrt{1 - \rho_{\boldsymbol{U}}}, \tag{10}$$

$$\left|\frac{dz(\alpha)}{d\alpha}\right| \leq 4\sqrt{2}b\sigma_{\boldsymbol{v}}\sigma_{\boldsymbol{U}}\log(24h/\delta)\sqrt{h}(\sqrt{1 - \rho_{\boldsymbol{v}}} + \sqrt{1 - \rho_{\boldsymbol{U}}}), \tag{11}$$

$$\left|\frac{d^2z(\alpha)}{d\alpha^2}\right| \leq 8b\sigma_{\boldsymbol{v}}\sigma_{\boldsymbol{U}}\log(4h/\delta)\sqrt{h}\sqrt{(1 - \max\{\rho_{\boldsymbol{U}}, \rho_{\boldsymbol{v}}\})}, \tag{12}$$

*where* $\rho_{\boldsymbol{v}}$ *and* $\rho_{\boldsymbol{U}}$ *refer to the mask ratios (the proportion of zeros in the mask) of masks* $\boldsymbol{M_v}$ *and* $\boldsymbol{M_U}$ *respectively.*

*Proof:* Let's first define $g_{\alpha}(\boldsymbol{x}) = (\alpha\boldsymbol{U} + (1 - \alpha)\boldsymbol{U'})\boldsymbol{x}$. Then we can express $z_{\boldsymbol{x}}(\alpha)$ as:

$$z_{\boldsymbol{x}}(\alpha) = (\alpha\boldsymbol{v} + (1 - \alpha)\boldsymbol{v'})^{\top}\sigma(g_{\alpha}(\boldsymbol{x})) - \alpha\boldsymbol{v}^{\top}\sigma(\boldsymbol{U}\boldsymbol{x}) - (1 - \alpha)\boldsymbol{v'}^{\top}\sigma(\boldsymbol{U'}\boldsymbol{x}). \tag{13}$$

The first derivative of $z_{\boldsymbol{x}}(\alpha)$ with respect to $\alpha$ will be:

$$\frac{dz_{\boldsymbol{x}}(\alpha)}{d\alpha} = (\boldsymbol{v} - \boldsymbol{v'})^{\top}\sigma(g_{\alpha}(\boldsymbol{x})) + (\alpha\boldsymbol{v} + (1 - \alpha)\boldsymbol{v'})^{\top}\sigma'(g_{\alpha}(\boldsymbol{x})) - \boldsymbol{v}^{\top}\sigma(\boldsymbol{U}\boldsymbol{x}) + \boldsymbol{v'}^{\top}\sigma(\boldsymbol{U'}\boldsymbol{x}). \tag{14}$$

The second derivative with respect to $\alpha$ will be:

$$\frac{d^2z_{\boldsymbol{x}}(\alpha)}{d\alpha^2} = 2(\boldsymbol{v} - \boldsymbol{v'})^{\top}\sigma'(g_{\alpha}(\boldsymbol{x})) + (\alpha\boldsymbol{v} + (1 - \alpha)\boldsymbol{v'})^{\top}\sigma''(g_{\alpha}(\boldsymbol{x})). \tag{15}$$

We also assume that the number of hidden neurons $h$ is sufficiently large for the convenience of analysis as (Entezari et al., 2022) and we use $\#\{\boldsymbol{M_U} = i\}$ and $\#\{\boldsymbol{M_v} = i\}$ denote the number of $i$ in $\boldsymbol{M_U}$ and $\boldsymbol{M_v}$ respectively, $i = 1, 2$. In the following proof, we will make use of Hoeffding's inequality for sub-Gaussian distributions. Here, we state it for reference: Let $X_1, \ldots, X_n$ be $n$ independent random variables such that $X_i \sim \text{sub-G}\left(0, \sigma^2\right)$. Then for any $\boldsymbol{a} = (a_1, ..., a_n) \in \mathbb{R}^n$, we have

$$\mathbb{P}\left[|\sum_{i=1}^{n} a_i X_i| > t\right] \leq 2\exp\left(-\frac{t^2}{2\sigma^2||a||_2^2}\right).$$

**1)** For the 0-order difference equation, we have

$$|z_{\boldsymbol{x}}(\alpha)| = \left|\alpha\boldsymbol{v}^{\top}\left[\sigma(g_{\alpha}(\boldsymbol{x})) - \sigma(\boldsymbol{U}\boldsymbol{x})\right] + (1 - \alpha)\boldsymbol{v'}^{\top}\left[\sigma(g_{\alpha}(\boldsymbol{x})) - \sigma(\boldsymbol{U'}\boldsymbol{x})\right]\right| \tag{16}$$

$$\leq \alpha\left|\boldsymbol{v}^{\top}\left[(\sigma(g_{\alpha}(\boldsymbol{x})) - \sigma(\boldsymbol{U}\boldsymbol{x})\right]\right| + (1 - \alpha)\left|\boldsymbol{v'}^{\top}\left[\sigma(g_{\alpha}(\boldsymbol{x})) - \sigma(\boldsymbol{U'}\boldsymbol{x})\right]\right|. \tag{17}$$

Then we bound the first term and the second term is bounded similarly due to symmetry. For the **concentration upper bound** of the first term of Equation 17, we use the Hoeffding's inequality for elements of $\boldsymbol{v}$, with probability $1 - \frac{\delta}{k}$

$$\alpha\left|\boldsymbol{v}^{\top}\left[(\sigma(g_{\alpha}(\boldsymbol{x})) - \sigma(\boldsymbol{U}\boldsymbol{x})\right]\right| \leq \alpha\sigma_v\sqrt{2\log(2k/\delta)}\|\sigma(g_{\alpha}(\boldsymbol{x})) - \sigma(\boldsymbol{M}\boldsymbol{x})\|_2 \tag{18}$$

$$\leq \alpha\sigma_v\sqrt{2\log(2k/\delta)}\|g_{\alpha}(\boldsymbol{x}) - \boldsymbol{M}\boldsymbol{x}\|_2 \tag{19}$$

$$= \alpha(1 - \alpha)\sigma_v\sqrt{2\log(2k/\delta)}\|(\boldsymbol{U'} - \boldsymbol{U})\boldsymbol{x}\|_2. \tag{20}$$

Equation 19 is due to the fact that the ReLU activation function satisfies the Lipschitz continuous condition with constant 1. For the item $\|(\boldsymbol{U} - \boldsymbol{U'})\boldsymbol{x}\|_2$, notice that $U_{ij} = U'_{ij}$ when $M_{\boldsymbol{U},ij} = 0$, and then take a union bound, with probability $1 - \frac{\delta}{k}$, we have

$$\|(\boldsymbol{U} - \boldsymbol{U'})\boldsymbol{x}\|_2 \leq \sqrt{\sum_{i=1}^{h}|[\boldsymbol{M}_{\boldsymbol{U},i:} \odot (\boldsymbol{U}_{i,:} - \boldsymbol{U'}_{i,:})]\boldsymbol{x}|^2} \tag{21}$$

$$= \sqrt{\sum_{i=1}^{h}|(\boldsymbol{U}_{i,:} - \boldsymbol{U'}_{i,:})(\boldsymbol{M}_{\boldsymbol{U},i:} \odot \boldsymbol{x})|^2} \tag{22}$$

$$\leq \sigma_{\boldsymbol{U}}\sqrt{\sum_{i=1}^{h}\|\boldsymbol{M}_{\boldsymbol{U},i:} \odot \boldsymbol{x}\|_2^2}\sqrt{4\log(2hk/\delta)}. \tag{23}$$

Then take a union bound choosing $k = 4$ (because the union bound is taken for 4 equations, Equation 20 and Equation 23 for the first and the second terms in Equation 17 respectively. Subsequent values of $k$ are determined with a similar method.), with probability $1 - \delta$ we have

$$|z_{\boldsymbol{x}}(\alpha)| < 4\sqrt{2}\alpha(1 - \alpha)\sigma_{\boldsymbol{v}}\sigma_{\boldsymbol{U}}\log(8h/\delta)\sqrt{\sum_{i=1}^{h}\|\boldsymbol{M}_{\boldsymbol{U},i:}\odot\boldsymbol{x}\|_2^2}. \tag{24}$$

Then integrate it on the region $\mathbb{X}$. With probability $1 - \delta$, we have

$$|z(\alpha)| \leq 4\sqrt{2}\alpha(1 - \alpha)\sigma_{\boldsymbol{v}}\sigma_{\boldsymbol{U}}\log(8h/\delta)b\sqrt{\frac{d}{d+2}}\sqrt{h - \frac{\#\{\boldsymbol{M}_{\boldsymbol{v}} = 0\}}{d}} \tag{25}$$

$$\leq \sqrt{2}\sigma_{\boldsymbol{v}}\sigma_{\boldsymbol{U}}\log(8h/\delta)b\sqrt{h - \frac{\#\{\boldsymbol{M}_{\boldsymbol{v}} = 0\}}{d}} \tag{26}$$

$$= \sqrt{2}\sigma_{\boldsymbol{v}}\sigma_{\boldsymbol{U}}\log(8h/\delta)b\sqrt{h}\sqrt{1 - \rho_{\boldsymbol{U}}}. \tag{27}$$

Equation 25 is due to fact that the integration $\frac{1}{|\mathbb{X}|}\int_{\mathbb{X}}\sqrt{\sum_{i=1}^{h}\|\boldsymbol{M}_{\boldsymbol{U},i:}\odot\boldsymbol{x}\|_2^2}d\boldsymbol{x}$ satisfies

$$\frac{1}{|\mathbb{X}|}\int_{\mathbb{X}}\sqrt{\sum_{i=1}^{h}\|\boldsymbol{M}_{\boldsymbol{U},i:}\odot\boldsymbol{x}\|_2^2}d\boldsymbol{x} \leq \sqrt{\left(\frac{1}{|\mathbb{X}|}\int_{\mathbb{X}}\sum_{i=1}^{h}\|\boldsymbol{M}_{\boldsymbol{U},i:}\odot\boldsymbol{x}\|_2^2 d\boldsymbol{x}\right)\left(\frac{1}{|\mathbb{X}|}\int_{\mathbb{X}}d\boldsymbol{x}\right)} \tag{28}$$

$$= \sqrt{\frac{1}{|\mathbb{X}|}\int_{\mathbb{X}}\#\{\boldsymbol{M}_{\boldsymbol{U}} = 1\}x_i^2 d\boldsymbol{x}} \tag{29}$$

$$= \sqrt{\frac{\#\{\boldsymbol{M}_{\boldsymbol{U}} = 1\}}{d}\frac{1}{|\mathbb{X}|}\int_{\mathbb{X}}\|\boldsymbol{x}\|_2^2 d\boldsymbol{x}} \tag{30}$$

$$= \sqrt{\left(h - \frac{\#\{\boldsymbol{M}_{\boldsymbol{U}} = 0\}}{d}\right)\frac{db^2}{d+2}}, \tag{31}$$

where Equation 28 is due to Cauchy-Schwarz inequality of integration, Equation 29 and Equation 30 is due to the symmetry of different components of $\boldsymbol{x}$ and Equation 31 is due to the integration $\frac{1}{|\mathbb{X}|}\int_{\mathbb{X}}\|\boldsymbol{x}\|_2^k d\boldsymbol{x} = \frac{db^k}{d+k}, k \in \mathbb{Z}$.

**2)** For the first derivative, we have

$$\left|\frac{dz_{\boldsymbol{x}}(\alpha)}{d\alpha}\right| \leq \left|(\boldsymbol{v} - \boldsymbol{v}')^\top\sigma(g_\alpha(\boldsymbol{x}))\right| + \left|(\alpha\boldsymbol{v} + (1 - \alpha)\boldsymbol{v}')^\top\sigma'(g_\alpha(\boldsymbol{x}))\right| + |v^\top\sigma(\boldsymbol{U}\boldsymbol{x}) - v'^\top\sigma(\boldsymbol{U}'\boldsymbol{x})|. \tag{32}$$

**i)** For the **concentration upper bound** of the first term of Equation 32, we use the Hoeffding's inequality for elements of $\boldsymbol{v} - \boldsymbol{v}'$ and notice that $v_i - v_i' = 0$ when $\boldsymbol{M}_{\boldsymbol{v},i} = 0$, with probability $1 - \frac{\delta}{k}$

$$\left|(\boldsymbol{v} - \boldsymbol{v}')^\top\sigma(g_\alpha(\boldsymbol{x}))\right| \leq \sigma_{\boldsymbol{v}}\sqrt{4\log(2k/\delta)}\|\boldsymbol{M}_{\boldsymbol{v}}\odot\sigma(g_\alpha(\boldsymbol{x}))\|_2 \tag{33}$$

$$\leq \sigma_{\boldsymbol{v}}\sqrt{4\log(2k/\delta)}\|\boldsymbol{M}_{\boldsymbol{v}}\odot g_\alpha(\boldsymbol{x})\|_2 \tag{34}$$

$$\leq \sigma_{\boldsymbol{v}}\sqrt{4\log(2k/\delta)}(\alpha\|\boldsymbol{M}_{\boldsymbol{v}}\odot\boldsymbol{U}\boldsymbol{x}\|_2 + (1 - \alpha)\|\boldsymbol{M}_{\boldsymbol{v}}\odot\boldsymbol{U}'\boldsymbol{x}\|_2). \tag{35}$$

Equation 34 is due to the property of ReLU activation function that $|\sigma(x)| < |x|$. The Hoffding's inequality is used again for each row $i$ of matrix $\boldsymbol{U}$ and $\boldsymbol{U}'$ with $\boldsymbol{M}_{\boldsymbol{v},i} = 1$, and after taking a union bound, we have the following inequality with probability $1 - \frac{\delta}{k}$,

$$\|\boldsymbol{M}_{\boldsymbol{v}}\odot\boldsymbol{U}\boldsymbol{x}\|_2 = \sqrt{\sum_{\boldsymbol{M}_{\boldsymbol{v},i} = 1}|\boldsymbol{U}_{i,:}\boldsymbol{x}|^2} \tag{36}$$

$$\leq \sigma_{\boldsymbol{U}}\sqrt{2(h - \#\{\boldsymbol{M}_{\boldsymbol{v}} = 0\})\log(2hk/\delta)}\|\boldsymbol{x}\|_2. \tag{37}$$

$\|\boldsymbol{M}_{\boldsymbol{v}}\odot\boldsymbol{U}'\boldsymbol{x}\|_2$ can be calculated similarly to Equation 37. Then after taking a union bound, with $1 - \frac{\delta}{k}$ the first term is bounded as

$$\left|(\boldsymbol{v} - \boldsymbol{v}')^\top\sigma(g_\alpha(\boldsymbol{x}))\right| \leq 2\sqrt{2}\sqrt{h - \#\{\boldsymbol{M}_{\boldsymbol{v}} = 0\}}\sigma_{\boldsymbol{v}}\sigma_{\boldsymbol{U}}\log(6hk/\delta)\|\boldsymbol{x}\|_2. \tag{38}$$

**ii)** For the **concentration upper bound** of the second term of Equation 32, we use the Hoeffeding's inequality for each element of $\boldsymbol{v}$ and $\boldsymbol{v}'$ and take a union bound, with probability $1 - \frac{\delta}{k}$ we have the following inequality,

$$
\begin{aligned}
& \left| (\alpha\boldsymbol{v} + (1-\alpha)\boldsymbol{v}')^\top \sigma'(g_\alpha(\boldsymbol{x})) \right| \\
&= \left| (\alpha\boldsymbol{v} + (1-\alpha)\boldsymbol{v}')^\top \sigma'(\boldsymbol{y})|_{\boldsymbol{y}=g_\alpha(\boldsymbol{x})} \odot (\boldsymbol{U} - \boldsymbol{U}')\boldsymbol{x}) \right| & (39) \\
&\leq \sqrt{\alpha^2 + (1-\alpha)^2}\sigma_{\boldsymbol{v}}\sqrt{2log(2k/\delta)} \|\sigma'(\boldsymbol{y})|_{\boldsymbol{y}=g_\alpha(\boldsymbol{x})} \odot (\boldsymbol{U} - \boldsymbol{U}')\boldsymbol{x})\|_2 & (40) \\
&\leq \sigma_{\boldsymbol{v}}\sqrt{log(2k/\delta)}\|(\boldsymbol{U} - \boldsymbol{U}')\boldsymbol{x}\|_2. & (41)
\end{aligned}
$$

Equation 39 is due to the chain rule of differentiation and Equation 40 is due to the fact that the property $|\sigma'(\cdot)| < 1$ of the ReLU activation function. The term $\|(\boldsymbol{U} - \boldsymbol{U}')\boldsymbol{x}\|_2 \leq \sigma_{\boldsymbol{U}}\sqrt{\sum_{i=1}^{h}\|\boldsymbol{M}_{\boldsymbol{U},i:} \odot \boldsymbol{x}\|_2^2}\sqrt{4\log(2hk/\delta)}$ is obtained in Equation 23. Then with $1 - \frac{\delta}{k}$ after taking a union bound, the second term is bounded as

$$
|(\alpha\boldsymbol{v} + (1-\alpha)\boldsymbol{v}')^\top \sigma'(g_\alpha(\boldsymbol{x})) \leq 2\log(4hk/\delta)\sigma_{\boldsymbol{v}}\sigma_{\boldsymbol{U}}\sqrt{\sum_{i=1}^{h}\|\boldsymbol{M}_{\boldsymbol{U},i:} \odot \boldsymbol{x}\|_2^2}. \quad (42)
$$

**iii)** For the **concentration upper bound** of the third term of Equation 32, first write it as

$$
\left| \boldsymbol{v}^\top\sigma(\boldsymbol{U}\boldsymbol{x}) - \boldsymbol{v}'^\top\sigma(\boldsymbol{U}'\boldsymbol{x}) \right| = \left| \boldsymbol{v}^\top\sigma(\boldsymbol{U}\boldsymbol{x}) - \boldsymbol{v}^\top\sigma(\boldsymbol{U}'\boldsymbol{x}) + \boldsymbol{v}^\top\sigma(\boldsymbol{U}'\boldsymbol{x}) - \boldsymbol{v}'^\top\sigma(\boldsymbol{U}'\boldsymbol{x}) \right| \quad (43)
$$

$$
\leq \left| \boldsymbol{v}^\top [\sigma(\boldsymbol{U}\boldsymbol{x}) - \sigma(\boldsymbol{U}'\boldsymbol{x})] \right| + \left| (\boldsymbol{v} - \boldsymbol{v}')^\top\sigma(\boldsymbol{U}'\boldsymbol{x}) \right|. \quad (44)
$$

Then we use the Hoeffeding's inequality for each element of $\boldsymbol{v}$ and $\boldsymbol{v}'$ and notice that $v_i - v_i' = 0$ when $M_{\boldsymbol{v},i} = 0$. After taking a union bound, with probability $1 - \frac{\delta}{k}$ we have the following inequality,

$$
\begin{aligned}
& \left| \boldsymbol{v}^\top\sigma(\boldsymbol{U}\boldsymbol{x}) - \boldsymbol{v}'^\top\sigma(\boldsymbol{U}'\boldsymbol{x}) \right| \\
&\leq \sigma_{\boldsymbol{v}}\sqrt{2\log(4k/\delta)}\|\sigma(\boldsymbol{U}\boldsymbol{x}) - \sigma(\boldsymbol{U}'\boldsymbol{x})\|_2 + \sigma_{\boldsymbol{v}}\sqrt{4\log(4k/\delta)}\|\boldsymbol{M}_{\boldsymbol{v}} \odot \sigma(\boldsymbol{U}'\boldsymbol{x})\|_2 & (45) \\
&\leq \sigma_{\boldsymbol{v}}\sqrt{2\log(4k/\delta)}\|(\boldsymbol{U} - \boldsymbol{U}')\boldsymbol{x})\|_2 + \sigma_{\boldsymbol{v}}\sqrt{4\log(4k/\delta)}\|\boldsymbol{M}_{\boldsymbol{v}} \odot \boldsymbol{U}'\boldsymbol{x}\|_2. & (46)
\end{aligned}
$$

Equation 46 is due to the fact the ReLU activation function $\sigma(\cdot)$ satisfied the Lipschitz continuity condition with constant 1 and $|\sigma(x)| \leq |x|$. The term $\|(\boldsymbol{U} - \boldsymbol{U}')\boldsymbol{x})\|_2 \leq \sigma_{\boldsymbol{U}}\sqrt{\sum_{i=1}^{h}\|\boldsymbol{M}_{\boldsymbol{U},i:} \odot \boldsymbol{x}\|_2^2}\sqrt{4\log(2hk/\delta)}$ in Equation 46 can be calculated as in Equation 23 with probability $1 - \frac{\delta}{k}$ and the term $\|\boldsymbol{M}_{\boldsymbol{v}} \odot \boldsymbol{U}'\boldsymbol{x}\|_2 \leq \sigma_{\boldsymbol{U}}\sqrt{2(h - \#\{\boldsymbol{M}_{\boldsymbol{v}} = 0\})\log(2hk/\delta)}\|\boldsymbol{x}\|_2$ can be caluclated as in Equation 37 with probability $1 - \frac{\delta}{k}$. Then take the union bound, with probability $1 - \frac{\delta}{k}$ we have

$$
\begin{aligned}
& \left| \boldsymbol{v}^\top\sigma(\boldsymbol{U}\boldsymbol{x}) - \boldsymbol{v}'^\top\sigma(\boldsymbol{U}'\boldsymbol{x}) \right| \\
&\leq \sigma_{\boldsymbol{v}}\sigma_{\boldsymbol{U}}\log(8kh/\delta)(2\sqrt{2}\sqrt{\sum_{i=1}^{h}\|\boldsymbol{M}_{\boldsymbol{U},i:} \odot \boldsymbol{x}\|_2^2} + 2\sqrt{2}\sqrt{h - \#\{\boldsymbol{M}_{\boldsymbol{v}} = 0\}}\|\boldsymbol{x}\|_2). & (47)
\end{aligned}
$$

In conjunction with analyses **i)**,**ii)** and **iii)** and take a union bound choosing $k = 3$, we have with probability $1 - \delta$,

$$\left|\frac{dz_{\boldsymbol{x}}(\alpha)}{d\alpha}\right| \leq \sqrt{h - \#\{\boldsymbol{M_v} = 0\}} 2\sqrt{2}\sigma_{\boldsymbol{v}}\sigma_{\boldsymbol{U}} \log(18h/\delta)\|\boldsymbol{x}\|_2$$

$$+ 2log(12h/\delta)\sigma_{\boldsymbol{v}}\sigma_{\boldsymbol{U}} \sqrt{\sum_{i=1}^{h} \|\boldsymbol{M_{U,i:}} \odot \boldsymbol{x}\|_2^2}$$

$$+ \sigma_{\boldsymbol{v}}\sigma_{\boldsymbol{U}} \log(24h/\delta)(2\sqrt{2}\sqrt{\sum_{i=1}^{h} \|\boldsymbol{M_{U,i:}} \odot \boldsymbol{x}\|_2^2} + 2\sqrt{2}\sqrt{h - \#\{\boldsymbol{M_v} = 0\}}\|\boldsymbol{x}\|_2) \quad (48)$$

$$\leq \sqrt{h - \#\{\boldsymbol{M_v} = 0\}} 4\sqrt{2}\sigma_{\boldsymbol{v}}\sigma_{\boldsymbol{U}} \log(24h/\delta)\|\boldsymbol{x}\|_2$$

$$+ 4\sqrt{2}log(24h/\delta)\sigma_{\boldsymbol{v}}\sigma_{\boldsymbol{U}} \sqrt{\sum_{i=1}^{h} \|\boldsymbol{M_{U,i:}} \odot \boldsymbol{x}\|_2^2}. \quad (49)$$

Then integrate them on the region $\mathbb{X}$. With probability $1 - \delta$ we have

$$\left|\frac{dz(\alpha)}{d\alpha}\right| \leq \sqrt{h - \#\{\boldsymbol{M_v} = 0\}} 4\sqrt{2}\sigma_{\boldsymbol{v}}\sigma_{\boldsymbol{U}} \log(24h/\delta)\frac{db}{d+1}$$

$$+ 4\sqrt{2}\sigma_{\boldsymbol{v}}\sigma_{\boldsymbol{U}} \log(24h/\delta)\sigma_{\boldsymbol{v}}\sigma_{\boldsymbol{U}} b\sqrt{\frac{d}{d+2}}\sqrt{h - \frac{\#\{\boldsymbol{M_U} = 0\}}{d}} \quad (50)$$

$$\leq 4\sqrt{2}b\sigma_{\boldsymbol{v}}\sigma_{\boldsymbol{U}} \log(24h/\delta)(\sqrt{h - \#\{\boldsymbol{M_v} = 0\}} + \sqrt{h - \frac{\#\{\boldsymbol{M_U} = 0\}}{d}}) \quad (51)$$

$$= 4\sqrt{2}b\sigma_{\boldsymbol{v}}\sigma_{\boldsymbol{U}}\sqrt{h} \log(24h/\delta)(\sqrt{1 - \rho_{\boldsymbol{v}}} + \sqrt{1 - \rho_{\boldsymbol{M}}}). \quad (52)$$

Equation 50 is due to the integration $\frac{1}{|\mathbb{X}|}\int_{\mathbb{X}} \|\boldsymbol{x}\|_2 d\boldsymbol{x} = \frac{db}{d+1}$ and $\frac{1}{|\mathbb{X}|}\int_{\mathbb{X}} \sqrt{\sum_{i=1}^{h} \|\boldsymbol{M_{U,i:}} \odot \boldsymbol{x}\|_2^2} d\boldsymbol{x} \leq \sqrt{(h - \frac{\#\{\boldsymbol{M_U} = 0\}}{d})\frac{db^2}{d+2}}$ from Equation 31

**3)** For the second derivative, we have

$$\left|\frac{d^2 z_{\boldsymbol{x}}(\alpha)}{d\alpha^2}\right| \leq 2\left|(\boldsymbol{v} - \boldsymbol{v}')^{\top}\sigma'(g_{\alpha}(\boldsymbol{x}))\right| + \left|(\alpha\boldsymbol{v} + (1 - \alpha)\boldsymbol{v}')^{\top}\sigma''(g_{\alpha}(\boldsymbol{x}))\right|. \quad (53)$$

**i)** For the **concentration upper bound** of the first term of Equation 53, we use the Hoeffding's inequality for each element of $\boldsymbol{v} - \boldsymbol{v}'$ and notice that $v_i - v_i' = 0$ when $M_{\boldsymbol{v},i} = 0$, with probability $1 - \frac{\delta}{k}$, we have

$$2\left|(\boldsymbol{v} - \boldsymbol{v}')^{\top}\sigma'(g_{\alpha}(\boldsymbol{x}))\right| = 2\left|(\boldsymbol{v} - \boldsymbol{v}')^{\top}\sigma'(\boldsymbol{y})|_{\boldsymbol{y}=g_{\alpha}(\boldsymbol{x})} \odot (\boldsymbol{U} - \boldsymbol{U}')\boldsymbol{x}\right| \quad (54)$$

$$= 2\left|(\boldsymbol{v} - \boldsymbol{v}')^{\top}\boldsymbol{M_v} \odot \sigma'(\boldsymbol{y})|_{\boldsymbol{y}=g_{\alpha}(\boldsymbol{x})} \odot (\boldsymbol{U} - \boldsymbol{U}')\boldsymbol{x}\right| \quad (55)$$

$$\leq 4\sigma_{\boldsymbol{v}}\sqrt{\log(2k/\delta)}\|\boldsymbol{M_v} \odot \sigma'(\boldsymbol{y})|_{\boldsymbol{y}=g_{\alpha}(\boldsymbol{x})} \odot (\boldsymbol{U} - \boldsymbol{U}')\boldsymbol{x}\|_2 \quad (56)$$

$$\leq 4\sigma_{\boldsymbol{v}}\sqrt{\log(2k/\delta)}\|\boldsymbol{M_v} \odot (\boldsymbol{U} - \boldsymbol{U}')\boldsymbol{x}\|_2. \quad (57)$$

Equation 54 is due to the chaine rule of differentiation, Equation 55 is due to $v_i = v_i'$ when $M_{\boldsymbol{v},i} = 0$, Equation 56 is due to Hoeffding's inequation and Equation 57 is due to the property $|\sigma'(x)| < 1$ of the ReLU activation function. For the item $\|\boldsymbol{M_v} \odot (\boldsymbol{U} - \boldsymbol{U}')\boldsymbol{x}\|_2$, notice that $U_{ij} = U_{ij}'$ when $M_{\boldsymbol{U},ij} = 0$ and take a union bound with probability $1 - \frac{\delta}{k}$, we have

$$\|\boldsymbol{M_v} \odot (\boldsymbol{U} - \boldsymbol{U}')\boldsymbol{x}\|_2 \leq \sqrt{\sum_{M_{\boldsymbol{v},i}=1} |[\boldsymbol{M_{U,i:}} \odot (\boldsymbol{U}_{i,:} - \boldsymbol{U}_{i,:}')]\boldsymbol{x}|^2} \quad (58)$$

$$\leq \sqrt{\sum_{M_{\boldsymbol{v},i}=1} |(\boldsymbol{U}_{i,:} - \boldsymbol{U}_{i,:}')(\boldsymbol{M_{U,i:}} \odot \boldsymbol{x})|^2} \quad (59)$$

$$\leq \sigma_{\boldsymbol{U}}\sqrt{\sum_{M_{\boldsymbol{v},i}=1} \|\boldsymbol{M_{U,i:}} \odot \boldsymbol{x}\|_2^2}\sqrt{4\log(2hk/\delta)}. \quad (60)$$

Then with $1 - \frac{\delta}{k}$ after taking a union bound, the first term is bounded as

$$2\left|(\boldsymbol{v} - \boldsymbol{v}')^\top \sigma'(g_\alpha(\boldsymbol{x}))\right| \le 8\sigma_{\boldsymbol{v}}\sigma_{\boldsymbol{U}} \log(4hk/\delta)\sqrt{\sum_{M_{\boldsymbol{v},i}=1} \|\boldsymbol{M}_{\boldsymbol{U},i:} \odot \boldsymbol{x}\|_2^2}. \tag{61}$$

**ii)** For the **concentration upper bound** of the second term of Equation 53, note that property $\sigma''(x) = 0$ of ReLU activation function, then

$$\left|(\alpha\boldsymbol{v} + (1-\alpha)\boldsymbol{v}')^\top \sigma''(g_\alpha(\boldsymbol{x}))\right|$$
$$= \left|(\alpha\boldsymbol{v} + (1-\alpha)\boldsymbol{v}')^\top \sigma''(\boldsymbol{y})|_{\boldsymbol{y}=g_\alpha(\boldsymbol{x})} \odot (\boldsymbol{U} - \boldsymbol{U}')\boldsymbol{x} \odot (\boldsymbol{U} - \boldsymbol{U}')\boldsymbol{x}\right| \tag{62}$$
$$= 0. \tag{63}$$

In conjunction with analyses **i)** and **ii)** and take a union bound choosing $k = 1$, with probability $1 - \delta$ we have

$$\left|\frac{d^2 z_{\boldsymbol{x}}(\alpha)}{d\alpha^2}\right| \le 8\sigma_{\boldsymbol{v}}\sigma_{\boldsymbol{U}} \log(4h/\delta)\sqrt{\sum_{M_{\boldsymbol{v},i}=1} \|\boldsymbol{M}_{\boldsymbol{U},i:} \odot \boldsymbol{x}\|_2^2}. \tag{64}$$

Then integrate them on the region $\mathbb{X}$. With probability $1 - \delta$ we have

$$\left|\frac{d^2 z(\alpha)}{d\alpha^2}\right| \le 8\sigma_{\boldsymbol{v}}\sigma_{\boldsymbol{U}} \log(4h/\delta)\sqrt{(h - \frac{\max\{\#\{\boldsymbol{M}_{\boldsymbol{U}} = 0\}, d\#\{\boldsymbol{M}_{\boldsymbol{v}} = 0\}\}}{d})\frac{db^2}{d+2}} \tag{65}$$

$$\le 8\sigma_{\boldsymbol{v}}\sigma_{\boldsymbol{U}} \log(4h/\delta)b\sqrt{(h - \frac{\max\{hd\rho_{\boldsymbol{U}}, hd\rho_{\boldsymbol{v}}\}}{d})} \tag{66}$$

$$\le 8\sigma_{\boldsymbol{v}}\sigma_{\boldsymbol{U}} \log(4h/\delta)b\sqrt{h}\sqrt{(1 - \max\{\rho_{\boldsymbol{U}}, \rho_{\boldsymbol{v}}\})}. \tag{67}$$

Equation 65 is due to the integration $\frac{1}{|\mathbb{X}|}\int_{\mathbb{X}}\sqrt{\sum_{i=1}^h \|\boldsymbol{M}_{\boldsymbol{U},i:} \odot \boldsymbol{x}\|_2^2}d\boldsymbol{x}$ satisfying

$$\frac{1}{|\mathbb{X}|}\int_{\mathbb{X}}\sqrt{\sum_{M_{\boldsymbol{v},i}=1} \|\boldsymbol{M}_{\boldsymbol{U},i:} \odot \boldsymbol{x}\|_2^2}d\boldsymbol{x} \le \sqrt{(\frac{1}{|\mathbb{X}|}\int_{\mathbb{X}}\sum_{M_{\boldsymbol{v},i}=1} \|\boldsymbol{M}_{\boldsymbol{U},i:} \odot \boldsymbol{x}\|_2^2 d\boldsymbol{x})(\frac{1}{|\mathbb{X}|}\int_{\mathbb{X}}d\boldsymbol{x})} \tag{68}$$

$$= \sqrt{\frac{1}{|\mathbb{X}|}\int_{\mathbb{X}}\#\{\boldsymbol{M}_{\boldsymbol{U}} \odot \boldsymbol{M}_{\boldsymbol{V}} = 1\}x_i^2 d\boldsymbol{x}} \tag{69}$$

$$= \sqrt{\frac{\#\{\boldsymbol{M}_{\boldsymbol{U}} \odot \boldsymbol{M}_{\boldsymbol{V}} = 1\}}{d}\frac{1}{|\mathbb{X}|}\int_{\mathbb{X}}\|x\|_2^2 d\boldsymbol{x}} \tag{70}$$

$$\le \sqrt{(h - \frac{\max\{\#\{\boldsymbol{M}_{\boldsymbol{U}} = 0\}, d\#\{\boldsymbol{M}_{\boldsymbol{v}} = 0\}\}}{d})\frac{db^2}{d+2}}, \tag{71}$$

where $\boldsymbol{M}_{\boldsymbol{V}}$ is the matrix whose each column is $\boldsymbol{M}_{\boldsymbol{v}}$. Equation 68 is due to Cauchy-Schwarz inequality of integration, Equation 69 and Equation 70 is due to the symmetry of different components of $\boldsymbol{x}$ and Equation 71 is due to the integration $\frac{1}{|\mathbb{X}|}\int_{\mathbb{X}}\|\boldsymbol{x}\|_2^2 d\boldsymbol{x} = \frac{db^2}{d+2}$ and $\#\{\boldsymbol{M}_{\boldsymbol{U}} \odot \boldsymbol{M}_{\boldsymbol{V}} = 1\} \le \min\{\#\{\boldsymbol{M}_{\boldsymbol{U}} = 1\}, \#\{\boldsymbol{M}_{\boldsymbol{V}} = 1\}\} = \min\{\#\{\boldsymbol{M}_{\boldsymbol{U}} = 1\}, d\#\{\boldsymbol{M}_{\boldsymbol{v}} = 1\}\}$. $\qquad\square$

## C   MORE ANALYSIS AND RESULTS

### C.1   MORE FORMS OF SUBSPACES REGARDING THE SUBSPACE HYPOTHESIS

We test LoRA (Hu et al., 2021) (a.k.a. learning in intrinsic dimension (Li et al., 2018a)) for our subspace hypothesis to see whether smaller subspaces will improve linear mode connectivity. LoRA is the abbreviation for Low-rank Adaption, and it is commonly used in parameter-efficient training of large language models (Hu et al., 2021). The idea of LoRA is to use a random low-rank matrix to map the parameter space into a subspace with lower dimensions.

Table 8: **Accuracy barriers of LoRA under different subspace dimensions.** The learning rates are the same as 0.01 and the dataset is CIFAR-10.

| Models\Dimensions | 2000 | 1000 | 500 |
|---|---|---|---|
| CNN | 0.148±0.052 | 0.325±0.11 | 0.428±0.023 |
| ResNet20 | 0.499±0.042 | 0.379±0.069 | 0.324±0.24 |

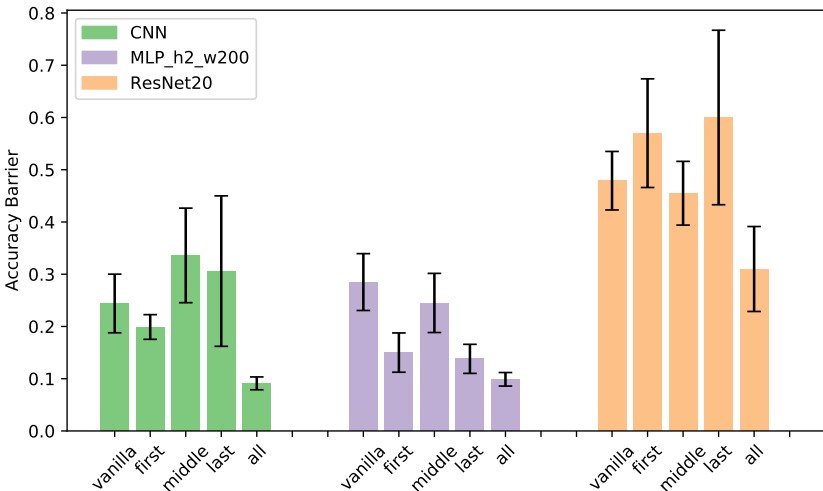

Figure 7: **Layer-wise analysis of TNA-PFN.** The dataset is CIFAR-10. "vanilla" refers to vanilla training. "first"/"middle"/"last" refers to only applying TNA-PFN to the first/middle/last layer. "all" refers to applying TNA-PFN to all layers (vanilla TNA-PFN). For CNN, the first layer is the convolution layer Conv2d(3, 32, 3), the middle layer is the convolution layer Conv2d(64, 64, 3), and the last layer is the fully connected layer Linear(64, 10) for classification; for MLP_h2_w200, the first layer is the fully connected layer Linear(32*32*3, 200), the middle layer is the fully connected layer Linear(200, 200), and the last layer is the fully connected layer Linear(200, 10) for classification; for ResNet20, the first layer is the convolution layer Conv2d(3, 16, kernel_size=3, stride=1, padding=1, bias=False), the middle layer is the middle block, and the last layer is the fully connected layer Linear(64*block.expansion, 10) for classification;

However, when we implement LoRA with random initialization, we find LoRA is very sensitive to the learning rate especially when the mapped dimension is low. Using the same learning rate, when we decrease the dimension, the network will become untrainable (with no generalization gains compared with random models). We hypothesize that the equivalent learning rates are different in different dimensions of LoRA, and for lower dimensions, the equivalent learning rates are larger. Thus, it is unfair to compare the connectivity of LoRAs under the same explicit learning rate since the learning rate is essential in determining the barriers (Adilova et al., 2023). However, we cannot quantify the relationship between the mapped dimension and the equivalent learning rate, and it is an interesting and open problem for future research.

Table 9: **Linear mode connectivity on Tiny ImageNet.** The $\rho$ for CNN is 0.4 and the $\rho$ for ResNet18 is 0.3. The learning rate is 0.08.

| Models | Metrics | TNA-PFN | Vanilla Train |
|--------|---------|---------|---------------|
| CNN | Avg. Acc. | $11.4 \pm 0.6$ | $9.85 \pm 0.3$ |
| | Interp. Acc. | $2.91 \pm 0.9$ | $1.4 \pm 0.2$ |
| | Acc. Barrier | $0.75 \pm 0.07$ (12.8% ↓) | $0.86 \pm 0.03$ |
| | Loss Barrier | $0.75 \pm 0.09$ (10.4% ↓) | $0.84 \pm 0.08$ |
| ResNet20 | Avg. Acc. | $31.6 \pm 0.4$ | $31.8 \pm 0.3$ |
| | Interp. Acc. | $12.5 \pm 2.1$ | $6.86 \pm 1.8$ |
| | Acc. Barrier | $0.60 \pm 0.07$ (23% ↓) | $0.78 \pm 0.06$ |
| | Loss Barrier | $1.2 \pm 0.09$ (22.2% ↓) | $1.6 \pm 0.2$ |

essential in determining the barriers (Adilova et al., 2023). However, we cannot quantify the relationship between the mapped dimension and the equivalent learning rate, and it is an interesting and open problem for future research.

We still provide the preliminary results in Table 8. Under the same learning rate, we measure the accuracy barriers when changing the mapped dimensions. It is found that different models reflect different tendencies. When the model is CNN, lower dimensions result in higher barriers, while for ResNet20, lower dimensions may indicate lower barriers. We note this result may not be rigorous enough to draw a conclusion, and further investigations in future work are needed.

### C.2 MORE RESULTS AND ILLUSTRATIONS IN LINEAR MODE CONNECTIVITY

**Results on large-scale dataset.** We conduct experiments on Tiny ImageNet (tin, Accessed: 2023), a subset of ImageNet (Deng et al., 2009), containing 100000 images of 200 classes (500 for each class) downsized to 64×64 colored images. Each class has 500 training images, 50 validation images, and 50 test images. The result is shown in Table 9. It can be seen that under large-scale datasets, TNA-PFN also can reduce the barriers in the linear mode connectivity.

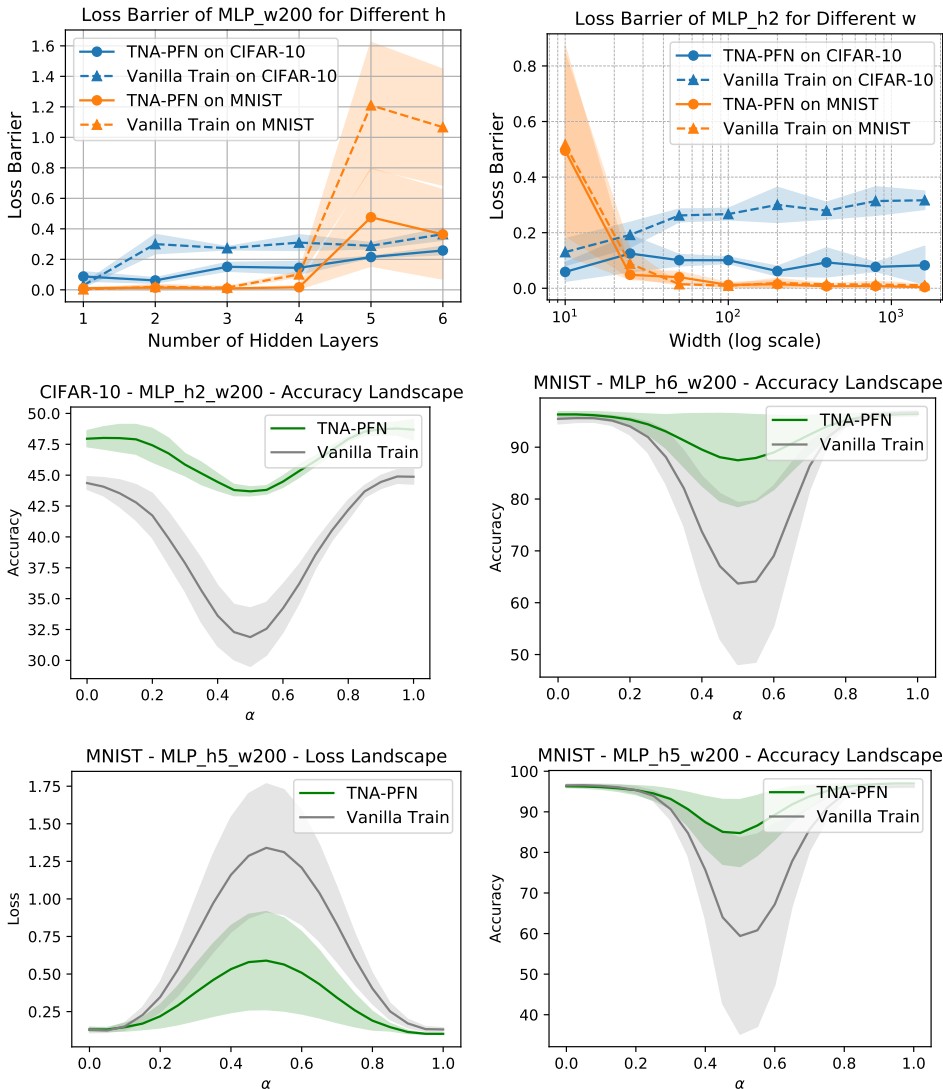

Figure 8: **Upper two: Loss barriers of MLP under different hidden layers ($h$) and widths ($w$). Middle two and Lower two: Accuracy and loss landscapes of MLPs.**

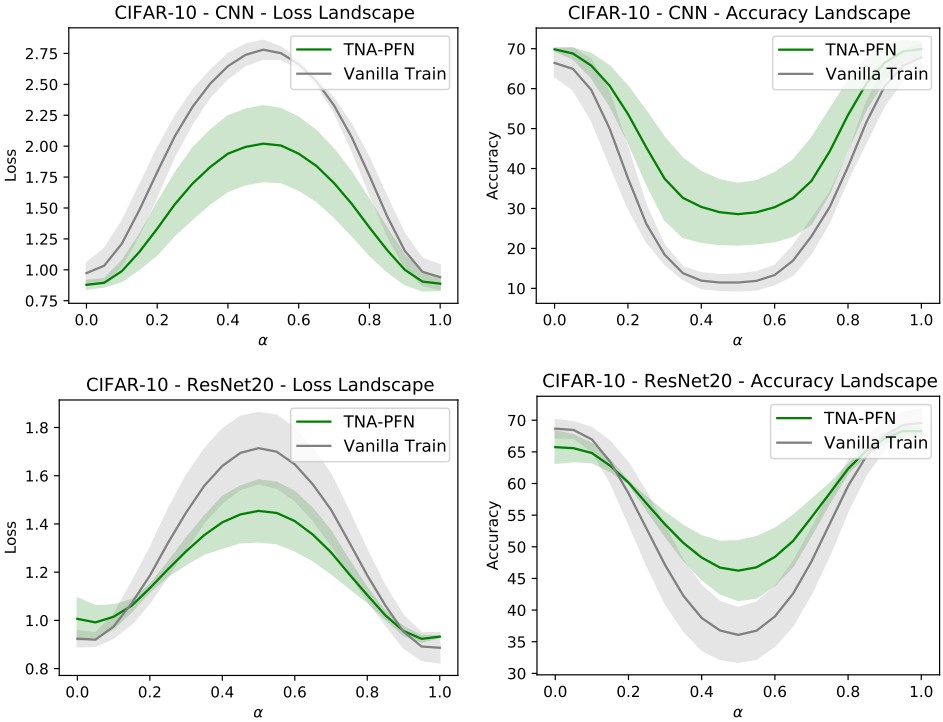

Figure 9: **Upper two: Loss and accuracy landscapes of CNN. Lower two: Loss and accuracy landscapes of ResNet20.**

**Layer-wise analysis.** We conduct a layer-wise analysis of TNA-PFN to see which layer matters most in improving LMC in Figure 7, and different model architecture poses different results. For simple CNN, only applying neuron fixing in the first layer (convolution) will improve LMC, and partially fixing weights in the middle (convolution) and the last (fully connected) layers will cause barrier increases. For MLP_h2_w200, we observe that independently fixing one layer will all cause barrier reductions, and the performance is more dominant when fixing the first and the last layers; jointly fixing all layers ("all") will have the lowest barrier. For ResNet20, it is revealed that only fixing the middle layers (the middle block) will cause barrier degradation.

**Extensions of Figure 3.** We provide more illustrations about the loss and accuracy barriers and landscapes in Figure 8.

**Extensions of Figure 4.** We provide illustrations about the loss and accuracy barriers of the Figure 4 results in Figure 9. It is obvious that TNA-PFN can lower the barriers in LMC.

**Loss and accuracy barriers w.r.t. epochs.** We demonstrate the barrier changes during training in Figure 10. It is shown that barriers increase during training, revealing that two independent networks diverge in parameter space. TNA-PFN has slower barrier-increasing rates than vanilla training.

**Extensions of Figure 2.** We provide more results about pruning and TNA-PFN under different mask ratios in Figure 11. Interestingly, for CNN, pruning and TNA-PFN improve both the accuracy and connectivity and the improvements go up along with the ratio increasing. On the other side, we observe an obvious accuracy-connectivity tradeoff for ResNet20 and it is more

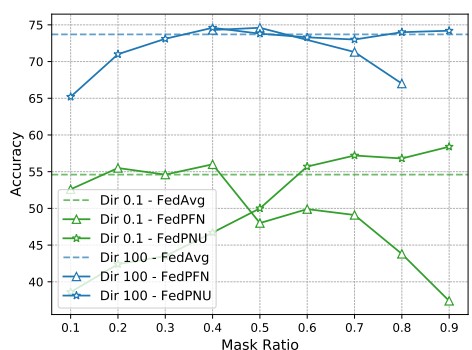

Figure 12: **Performances of FedPFN and FedPNU under different mask ratios.** CIFAR-10, ResNet20, and $E = 3$.

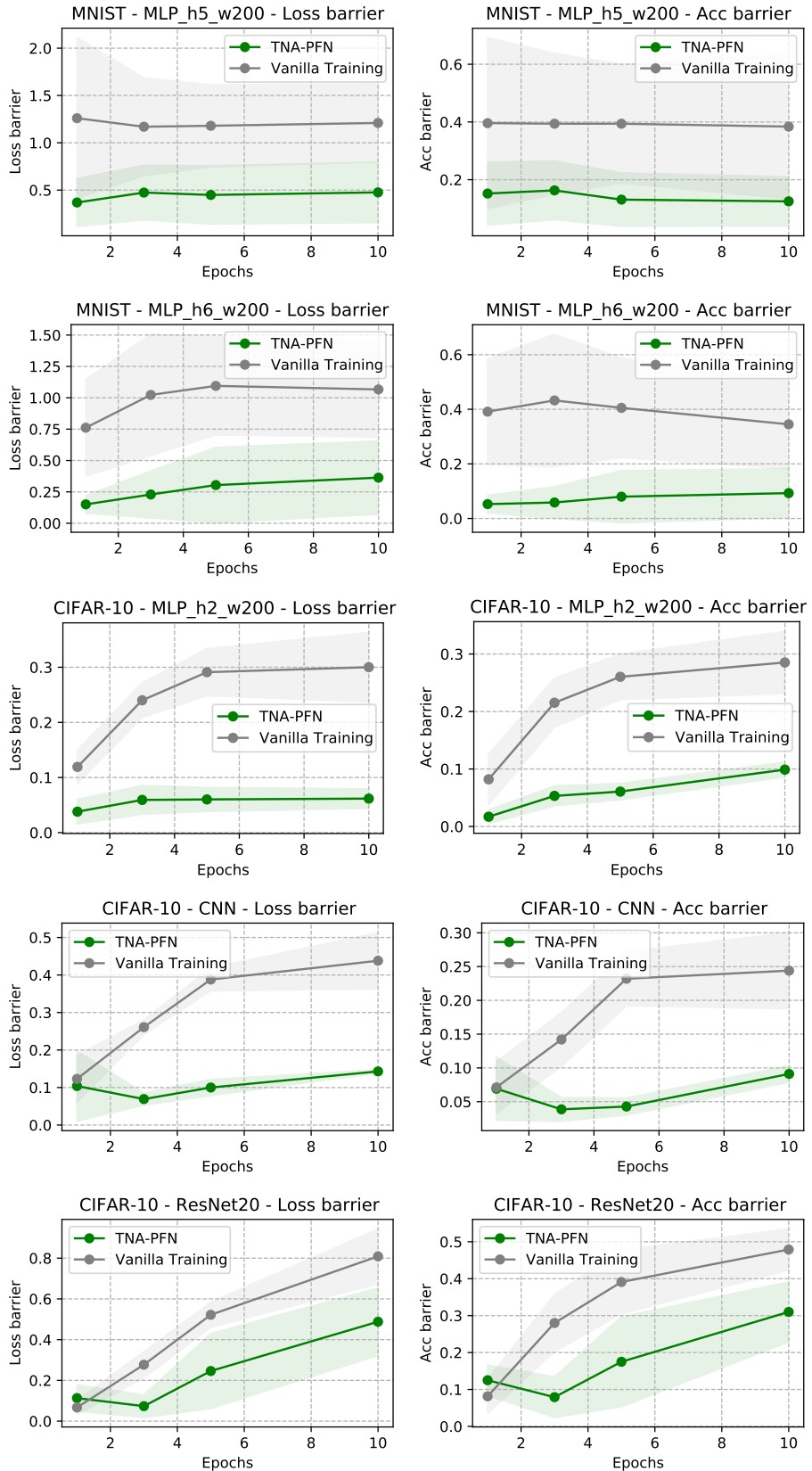

Figure 10: **Barrier changes during training for different datasets and models.**

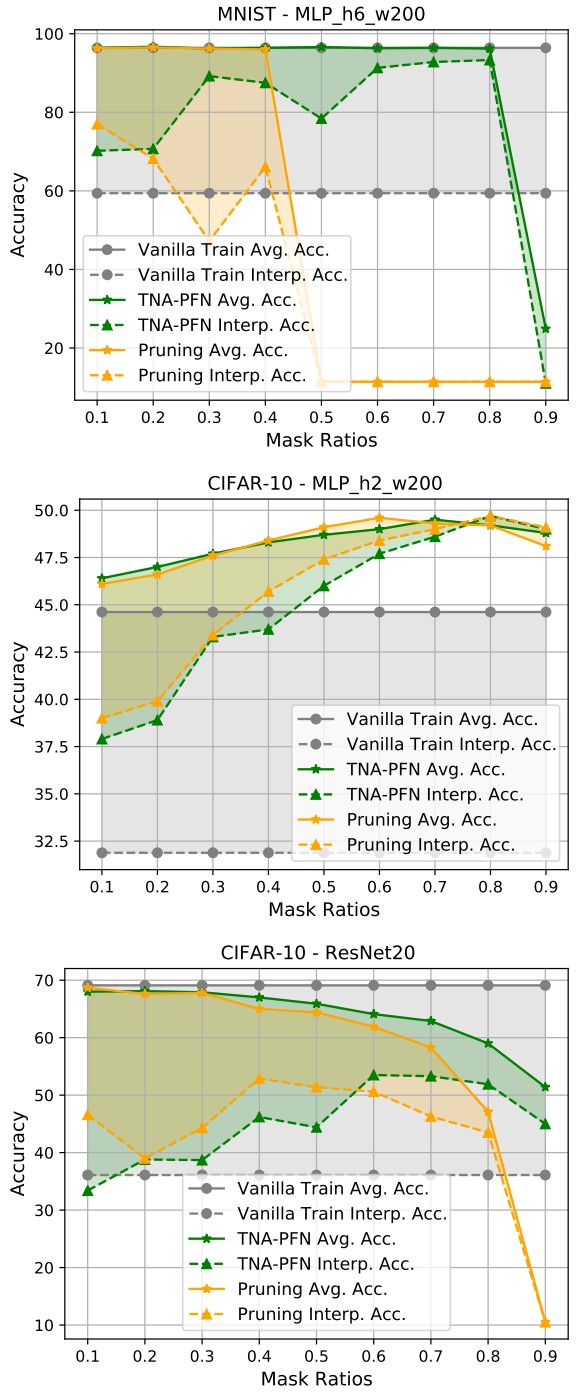

Figure 11: **More results about pruning and TNA-PFN under different mask ratios.** The shadow areas mean the accuracy barriers.

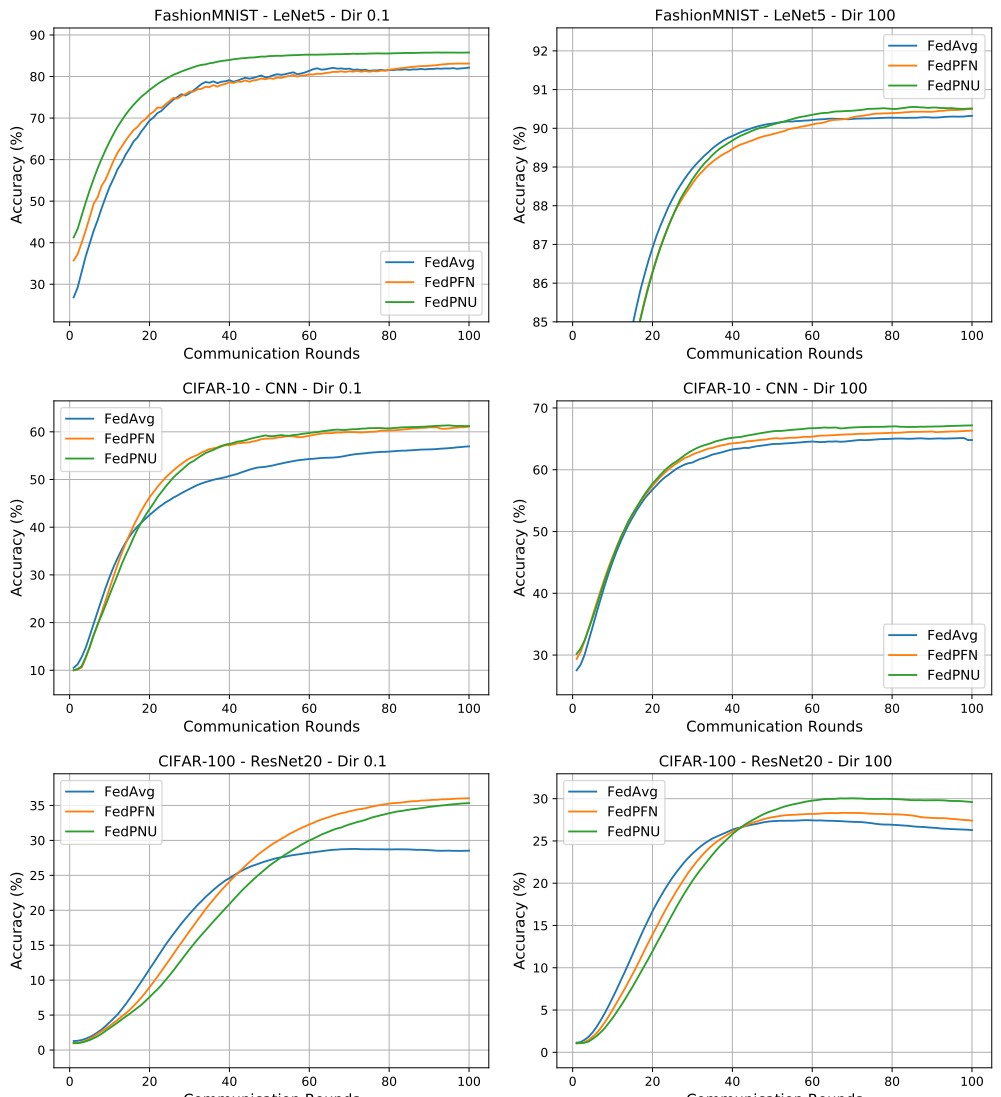

Figure 13: **Test accuracy curves of the federated learning methods.**

severe for pruning. Also, considering the layer-wise evaluation for ResNet in Figure 7, we reckon it is important to devise different mask strategies for the layers in ResNet and other deeper or more complex models.

### C.3 MORE RESULTS AND ILLUSTRATIONS IN FEDERATED LEARNING

**Extensions of Figure 6.** We show the performances of FedPFN and FedPNU under different mask ratios for ResNet20 in Figure 12. The results indicate that FedPFN is sensitive to the mask ratio while FedPNU is more robust. FedPNU reaches higher performances under higher mask ratios ($\rho \in [0.8, 0.9]$).

**Illustrations of the learning curves.** We present the test accuracy curves of FedAvg, FedPFN, and FedPNU in Figure 13. Our methods show dominant advantages over FedAvg in both IID and non-IID settings, especially for the more complex datasets, CIFAR-10 and CIFAR-100.

## D PRELIMINARY OF FEDERATED LEARNING

Federated learning usually involves a server and $n$ clients to jointly learn a global model without data sharing, which is originally proposed in (McMahan et al., 2017). Denote the set of clients by $\mathcal{S}$, the

labeled data of client $i$ by $\mathcal{D}_i = \{(x_j, y_j)\}_{j=1}^{N_i}$, and the parameters of the current global model by $\mathbf{w}_g^t$. FL starts with **client training** in parallel, initializing each clients' model $\mathbf{w}_i^t$ with $\mathbf{w}_g^t$.

FL is more communication-efficient than the conventional distributed training, that it assumes the clients train the models for epochs (the full data) instead of iterations (the mini-batch data) between the communications to the server. **The number of local epochs is denoted as $E$.**

In each local epoch, clients conduct SGD update with a local learning rate $\eta_l$, each SGD iteration shows as

$$\text{Client training:} \qquad \mathbf{w}_i^t \leftarrow \mathbf{w}_i^t - \eta_l \nabla \ell(B_k, \mathbf{w}_i^t), \text{ for } k = 1, 2, \cdots, K, \tag{72}$$

where $\ell$ is the loss function and $B_k$ is the mini-batch sampled from $\mathcal{D}_i$ at the $k$th iteration. After the client local updates, the server samples $m$ clients for aggregation. The client $i$'s pseudo gradient of local updates is denoted as $\mathbf{g}_i^t = \mathbf{w}_g^t - \mathbf{w}_i^t$. Then, the server conducts FEDAVG to aggregate the local updates into a new global model.

$$\text{Weighted Model aggregation:} \qquad \mathbf{w}_g^{t+1} = \mathbf{w}_g^t - \sum_{i=1}^{m} \lambda_i \mathbf{g}_i^t, \ \lambda_i = \frac{|\mathcal{D}_i|}{|\mathcal{D}|}, \forall i \in [m]. \tag{73}$$

With the updated global model $\mathbf{w}_g^{t+1}$, it then starts the next round of **client training**. The procedure of FL therefore iterates between Equation 72 and Equation 73, **for $T$ communication rounds**.

We assume the sum of clients' data as $\mathcal{D} = \bigcup_{i \in \mathcal{S}} \mathcal{D}_i$. The IID data distributions of clients refer to that each client's distribution $\mathcal{D}_i$ is IID sampled from $\mathcal{D}$. However, in practical FL scenarios, **heterogeneity** exists among clients that their data are **non-IID** with each other. Each client may have different data distributions in the input (e.g. image distribution) or output (e.g. label distribution).

## E   MORE DISCUSSIONS ABOUT RELATED WORKS

In this section, we give a more detailed discussion of the related works.

**Relation with Frankle et al. (2020).** In Frankle et al. (2020), the authors use linear mode connectivity to study the performances of lottery-ticket-hypothesis-based pruning and find that the sparse pruned model with good connectivity will be more likely to reach the full accuracy after pruning. While in this paper, we find random pruning (not necessarily lottery tickets) can improve linear mode connectivity. Though the two papers both discuss the relationship between pruning and linear mode connectivity, they have different focuses and contributions: Frankle et al. (2020) finds LMC indicates better results of pruning, whereas we find pruning can improve LMC, and the causal logic is different.

**Subspace Learning.** Subspace learning has various forms, and we summarize other forms of subspaces that have not appeared in this paper. Intrinsic dimensions use a random low-rank matrix to map the network parameters into a subspace and it finds neural networks have lower intrinsic dimensions than the original dimension to reach a close or same accuracy (Li et al., 2018a). Further, LoRA introduces intrinsic dimension in parameter-efficient finetuning of large language models (Hu et al., 2021); and Gressmann et al. (2020) improves learning in intrinsic dimensions above random matrices. Additionally, subspaces are used in improving continual and incremental learning. Chaudhry et al. (2020) use a projection matrix to map the features into orthogonal subspaces to prevent catastrophic forgetting in continual learning; Akyürek et al. (2021) adds a subspace regularizer for improving few-shot class incremental learning.

**Subspace and Partial Training in Federated Learning.** Previous works in federated learning apply subspace or partial training methods, but they have different motivations and are orthogonal to our approaches FedPFN and FedPNU. In Isik et al. (2022), the authors propose to train a mask on a random network instead of training the neurons from communication efficiency; while in Li et al. (2021), it is proposed to learn a personalized sparsed network at clients. Additionally, partial training is adopted in federated learning to save computation and communication. Lee et al. (2023) proposes to train $\frac{1}{E}$ part of models and then aggregate on the server to relieve the negative effects brought by large local epochs. Yang et al. (2022) proposes each client train $\frac{1}{n}$ disjoint part of the whole model and combines the model on the server for efficiency. Niu et al. (2022) aims to enable large-model training at edges by decoupling the model into several principle sub-models. In Hahn et al. (2022), the authors utilize linear mode connectivity to improve personalization in federated learning.

**Loss Landscape of Neural Networks and Generalization.** Deep neural networks (DNNs) are highly non-convex and over-parameterized, and visualizing the loss landscape of DNNs (Li et al., 2018b; Vlaar & Frankle, 2022) helps understand the training process and the properties of minima.

There are mainly two lines of works about the loss landscape of DNNs. The first one is the linear interpolation of neural network loss landscape (Vlaar & Frankle, 2022; Garipov et al., 2018; Draxler et al., 2018), it plots linear slices of the landscape between two networks. In linear interpolation loss landscape, mode connectivity (Draxler et al., 2018; Vlaar & Frankle, 2022; Entezari et al., 2022) is referred to as the phenomenon that there might be increasing loss on the linear path between two minima found by SGD, and the loss increase on the path between two minima is referred to as (energy) barrier. It is also found that there may exist barriers between the initial model and the trained model (Vlaar & Frankle, 2022). The second line concerns the loss landscape around a trained model's parameters (Li et al., 2018b). It is shown that the flatness of loss landscape curvature can reflect the generalization (Foret et al., 2020; Izmailov et al., 2018) and top hessian eigenvalues can present flatness (Yao et al., 2020; Jastrzębski et al., 2018). Networks with small top hessian eigenvalues have flat curvature and generalize well. Previous works seek flatter minima for improving generalization by implicitly regularizing the hessian (Foret et al., 2020; Kwon et al., 2021; Du et al., 2021).

