# OpenReview forum: "Training-time Neuron Alignment for Improving Linear Mode Connectivity and Model Fusion"
_ICLR.cc/2024/Conference — Submitted to ICLR 2024_

### Official Review · Reviewer_kfmS · 2023-10-30

**Soundness:** 2 fair
**Presentation:** 2 fair
**Contribution:** 2 fair
**Rating:** 6
**Confidence:** 2

**Summary:**

The authors study the problem linear mode connectivity(LMC), starting from the assumption that the optimal network weights lie in a lower dimensional subspace. To measure LMC they use the commonly used Accuracy Barrier as a metric. They study the setting where a subset of fixed network weights is frozen during training, which leads to better LMC compared to vanilla training for the case of a single model as well as the case of multi-model fusion.
They also consider the setting of federated learning where at each time step the server freezes a subset of networks weights of all the clients.

**Strengths:**

- The method seems to outperform vanilla pruning  in Figure 2
- In the setting of federated learning the proposed have similar performance to previous baselines and and some cases even has better performance

**Weaknesses:**

- It is a bit hard to judge from the experimental results for what networks/settings one would prefer this method over other methods
- Lack of experiments at scale (e.g. ImageNet)

**Questions:**

- The method is conjectured to perform better for wider networks and worse for deeper networks. It would be interesting to have a more quantitative result  i.e. plot the performance for varying width and depth.
- For the setting of a single network and multi-model fusion why is there only a comparison to vanilla training and not other methods for improving LMC?
- Why is the accuracy not reported in Figure 1?
- Could the authors elaborate a bit how pruning for mask ratio 0.7 in Figure 2 causes the model to be as bad as random guessing? Do they expect similar behavior for larger datasets and models?

---

> ### Author Response · Authors · 2023-11-18
> **Response to Reviewer kfmS (1/2)**
>
> We appreciate the valuable reviews. We accordingly respond to the reviewer's comments below.
>
> > 1. Response to "It is a bit hard to judge from the experimental results for what networks/settings one would prefer this method over other methods."
> >
>
> Thank you for the comment. We would like clarify that our main contribution may lie in the new perspective and insights regarding improving training-time linear mode connectivity (instead of the post-training). The TNA-PFN is just a showcase and the verification of our hypothesis/perspective.
>
> Moreover, TNA-PFN has practical implications in the following aspects:
> - a. _Enable efficient re-basin and improve re-basin._ We note that in some cases, TNA-PFN + weight matching (WM) can achieve even better interpolated generalization compared with WM alone (Table 1 of the paper: MLP_h2_w200 for CIFAR-10 and MLP_h5_w200 for MNIST); also, it costs less computation of WM.
> - b. _Extensions to federated learning._ We extend TNA-PFN in federated learning to improve the generalization of the global model after model fusion. The proposed FedPFN and FedPNU are light-weight and effective.
>
> > 2. Response to "Lack of experiments at scale (e.g. ImageNet)."
> >
>
> Thanks for this helpful comment. Due to the time constraint of rebuttal period, we may not be able to implement the experiments on full ImageNet, as an alternative, we have conducted experiments on Tiny ImageNet [2], a subset of ImageNet.
>
> Tiny ImageNet contains 100000 64×64 colored images of 200 classes (500 for each class), which can be regarded as more large-scale. The results are in the following Table A, also added to the revised paper. It is observed that TNA-PFN can also reduce the barriers on Tiny ImageNet by up to 23%.
>
> **Table A. Linear mode connectivity on Tiny ImageNet.** The $\rho$ for CNN is 0.4 and the $\rho$ for ResNet18 is 0.3. The learning rate is 0.08.
>
> | **Models** | **Metrics**    | **TNA-PFN**                        | **Vanilla Train** |
> |------------|----------------|------------------------------------|-------------------|
> | CNN        | Avg. Acc.      | $11.4\pm0.6$                   | $9.85\pm0.3$   |
> |            | Interp. Acc.   | $2.91\pm0.9$                   | $1.4\pm0.2$     |
> |            | Acc. Barrier   | $0.75\pm0.07$ (12.8%↓)         | $0.86\pm 0.03$  |
> |            | Loss Barrier   | $0.75\pm0.09$ (10.4%↓)         | $0.84\pm0.08$   |
> | ResNet20   | Avg. Acc.      | $31.6\pm0.4$                   | $31.8\pm0.3$    |
> |            | Interp. Acc.   | $12.5\pm2.1$                   | $6.86\pm1.8$   |
> |            | Acc. Barrier   | $0.60\pm0.07$ (23%↓)           | $0.78\pm0.06$   |
> |            | Loss Barrier   | $1.2\pm0.09$ (22.2%↓)          | $1.6\pm0.2$     |
>
> > 3. Response to "The method is conjectured to perform better for wider networks and worse for deeper networks. It would be interesting to have a more quantitative result i.e. plot the performance for varying width and depth."
> >
>
> Thank you for the comment. Acutally, in Figures 3 and 8 of the initial submission, we have conducted experiments regarding the performances for varying width and depth. It seems that our method works well under different depths and widths.

---

> ### Author Response · Authors · 2023-11-18
> **Response to Reviewer kfmS (2/2)**
>
> > 4. Response to "For the setting of a single network and multi-model fusion why is there only a comparison to vanilla training and not other methods for improving LMC?"
> >
>
> Since it might be the first paper to improve linear mode connectivity (LMC) from the training time and previous methods are post-training, other methods are orthogonal to ours, as a result, we only compare with the vanilla training. But we have included experiments of previous post-hoc matching methods in Table 1 of the paper, showing our method is compatible with other post-hoc methods and can have advantages in efficiency.
>
> > 5. Response to "Why is the accuracy not reported in Figure 1?"
> >
>
> Thank you for the comment. We note that Figure 1 is just a toy illustration of how TNA-PFN might work, and it is not an experimental result, therefore, no accuracy is needed in Figure 1.
>
> > 6. Response to "Could the authors elaborate a bit how pruning for mask ratio 0.7 in Figure 2 causes the model to be as bad as random guessing? Do they expect similar behavior for larger datasets and models?"
> >
>
> Thanks for the comment. For more datasets and models, the results can be found in Figure 11 of the appendix. Also, more results and similar observations can be found in [1].
>
> Pruning sets the weights to zeros, and a high pruning ratio can lead to the deaths of a substantial proportion of neurons, missing important data features. Using the first layer as an example, setting half of the input weights to zero is akin to losing half of the initial data features. However, in the case of TNA-PFN, by fixing these weights, the features remain intact and are conveyed to the second layer. Consequently, excessive pruning can degrade the model's performance to the level of random guessing.
>
>
> ---
> [1] Fladmark E, Sajjad M H, Justesen L B. Exploring the Performance of Pruning Methods in Neural Networks: An Empirical Study of the Lottery Ticket Hypothesis[J]. arXiv preprint arXiv:2303.15479, 2023.

---

> ### Author Response · Authors · 2023-11-22
> **To Reviewer kfmS: hoping that our response could address your concerns**
>
> Dear Reviewer kfmS,
>
> Many thanks for your valuable comments. We have faithfully given detailed responses to your concerns during the rebuttal.
>
> Specifically, we address the following points:
>
> - We have conducted experiments at scale, on a subset of ImageNet, showing that TNA-PFN is also effective in reducing the barriers.
> - We have detailedly answered some questions raised by the reviewer.
>
> We would appreciate it if you could let us know if our response has sufficiently addressed your questions and thus kindly reconsider your score.
>
> Thank you.
>
> Best wishes,
>
> Authors

---

### Official Review · Reviewer_M3kr · 2023-10-30

**Soundness:** 3 good
**Presentation:** 3 good
**Contribution:** 2 fair
**Rating:** 5
**Confidence:** 3

**Summary:**

When training a neural network with SGD, different solutions in the parameter space can be obtained. The linear connection between two different solutions is called Linear Mode Connectivity (LMC). One commonly used approach to achieve LMC is to align the neurons of two network parameters through parameter permutation after training so that they are in the same loss basin. However, the number of permutation matrices is very large, and it requires a lot of computation because it is post-hoc. Therefore, the authors propose Training-time Neuron Alignment by Partially Fixing Neurons (TNA-PFN), which can align neurons at training time to create LMC. TNA-PFN is a method that learns in the parameter subspace by fixing part of the network parameters as initialization, based on the hypothesis that learning a network in an effective subspace with less permutation symmetry can lower the LMC barrier between trained networks. The authors support this hypothesis through both theoretical and experimental results. Also, the authors propose two algorithms, FedPFN and FedPNU, which adapt TNA-PFN to federate learning, and show that they have the potential to improve model fusion on different datasets.

**Strengths:**

I think the strengths are the simplicity of the method, the theoretical support, and the clean writing.
- The paper presents a new method to achieve LMC, and the idea of fixing a parameter to reduce permutation symmetry is novel.
- The paper is well written and easy to follow.
- The hypothesis and theoretical explanation are convincing.
- Experiments are extensive and show that TNA-PFN works in some practical applications.
- The paper proposes FedPFN and FedPNU, which can be used practically in Federated Learning, and shows that they work well in practice.

**Weaknesses:**

The biggest weakness seems to be the experimental results. The results are not promising enough to accept that TNA-PFN works effectively.

- As mentioned in the paper, the MNIST results in the second plot of Figure 3 and the experimental results of Entezari, et al. (2022) show that the barrier actually decreases as the network width increases. According to the theorem presented in the paper, the barrier may decrease as the learned dimension decreases, but it does not apply well to these experiments.
- In many experiments, the absolute barrier is too high when using TNA-PFN alone to be an LMC. Of course, it lowers the barrier compared to the vanilla train with no training, but many experiments show a non-negligible barrier.
- There is almost no difference in LMC performance between weight matching (WM) after training with TNA-PFN and directly using weight matching. Weight matching after TNA-PFN requires a few fewer iterations, but I don't know how much of a cost savings this provides. From a practical perspective, it introduces an additional hyperparameter, the mask ratio $\rho$, so I'm not sure how much benefit there is compared to the cost of tuning it.
- It's good to have a variety of experiments, but there are no experiments on large datasets like ImageNet. It would have been nice to see some experiments on larger datasets, as they generally have different characteristics than MNIST and CIFAR10.
---
**Entezari, et al.** [The Role of Permutation Invariance in Linear Mode Connectivity of Neural Networks](https://openreview.net/pdf?id=dNigytemkL). *ICLR*, 2022

**Questions:**

- In the experiments in the paper, the mask ratio was set to 0.4; was this an experimentally tuned value, or is there some underlying theoretical basis?
- Which of the three algorithms presented in Ainsworth et al. (2023) was used for weight matching (WM)? What is the approximate computational cost per iteration?
- Do FedPFN or FedPNU work in general learning situations other than federated learning, and can they be used for model ensembling?
- I'm not sure why the LoRA section was added to Appendix C.1 and what it is trying to say.
---
**Ainsworth et al.** [Git Re-Basin: Merging Models modulo Permutation Symmetries](https://openreview.net/forum?id=CQsmMYmlP5T). *ICLR*, 2023

---

> ### Author Response · Authors · 2023-11-18
> **Response to Reviewer M3kr (1/3)**
>
> We appreciate the valuable reviews. We accordingly respond to the reviewer's comments below.
>
> > 1. Response to "The biggest weakness seems to be the experimental results. The results are not promising enough to accept that TNA-PFN works effectively."
> >
>
> Thanks for this comment. We would like to kindly clarify our contributions.
> - **1)** *New insights into linear mode connectivity and permutation invariance from training time.* Our novelty lies in the **training-time** "subspace hypothesis" perspective while previous works only focus on **post-training** connectivity via permutations symmetries (re-basin). The proposed TNA-PFN is just a showcase and verification of the hypothesis/perspective. The contributions are alike [1] in that its main contribution is the hypothesis of linear mode connectivity and permutation invariance and the simulated annealing algorithm (which may not be very effective) is just the verification. We believe our new perspective can inspire more future work in the community.
> - **2)** *The applications of training-time neuron alignment in federated learning.* While linear mode connectivity is somewhat theoretical, we propose two algorithms in federated learning to showcase how improving connectivity during training can help practices. The proposed methods are lightweight and effective.
>
>
> > 2. Response to "As mentioned in the paper, the MNIST results in the second plot of Figure 3 and the experimental results of Entezari, et al. (2022) show that the barrier actually decreases as the network width increases. According to the theorem presented in the paper, the barrier may decrease as the learned dimension decreases, but it does not apply well to these experiments."
> >
>
> We apologize that the previous theorem may cause potential misleadings and we have refined the theorem to make it more solid.
>
> - The previous theorem may cause a misleading that lowering the network dimension by changing the network architecture can improve linear mode connectivity. Actually, this is not what we try to convey and is also inconsistent with the empirical observations.
> - After our revision, the revised theorem is more aligned with our hypothesis and method. It is proved that given a network architecture (without changing the dimension, i.e., width and depth), if we fix a proportion of parameters and train the remaining (like what TNA-PFN does), the linear mode connectivity of two models after training would be improved.
> - The revised theorem is Theorem 3.2 in blue on page 3, and the detailed proof is in the appendix. We hope the improved theorem can relieve the concerns and the potential misleading.
>
> > 3. Response to "In many experiments, the absolute barrier is too high when using TNA-PFN alone to be an LMC. Of course, it lowers the barrier compared to the vanilla train with no training, but many experiments show a non-negligible barrier."
> >
>
> Thanks for your comment. Actually, we didn't claim that we have completely _eliminated_ the barriers, and what we said is we largely _decreased_ the barriers. To the best of our knowledge, we might be the first to attempt to discover the potential of improving linear mode connectivity (LMC) during training, which is very difficult and challenging.
>
> SGD randomness results in SGD noise, and as a consequence, causes an LMC barrier after training. We make the preliminary attempt to fix a proportion of weights to reduce the permutation effects of SGD noise. However, it is impossible to eliminate the SGD noise when the model is independently trained, and as long as SGD noise remains, the LMC barrier will not be completely eliminated during training.
>
> We note even if the barrier may still exist after applying TNA-PFN, we have already achieved up to 70% reduction during training time, and it is promising enough for promoting the theoretical and empirical studies.

---

> ### Author Response · Authors · 2023-11-18
> **Response to Reviewer M3kr (2/3)**
>
> > 4. Response to "There is almost no difference in LMC performance between weight matching (WM) after training with TNA-PFN and directly using weight matching. Weight matching after TNA-PFN requires a few fewer iterations, but I don't know how much of a cost savings this provides. From a practical perspective, it introduces an additional hyperparameter, the mask ratio $\rho$, so I'm not sure how much benefit there is compared to the cost of tuning it."
> >
>
> We note that in some cases, TNA-PFN + weight matching (WM) can achieve even better interpolated generalization compared with WM alone (Table 1 of the paper: MLP_h2_w200 for CIFAR-10 and MLP_h5_w200 for MNIST); also, it costs less computation of WM.
>
> For the hyperparameter $\rho$, it can be seen from Figures 1 and 11 that TNA-PFN is not sensitive in terms of $\rho$. TNA-PFN could be effetive to improve the interpolated accuracy under a wide range of $\rho$, mainly from 0.1 to 0.6. Therefore, tuning $\rho$ may be not essential and the cost is marginal.
>
> > 5. Response to "It's good to have a variety of experiments, but there are no experiments on large datasets like ImageNet. It would have been nice to see some experiments on larger datasets, as they generally have different characteristics than MNIST and CIFAR10."
> >
>
> Thanks for this helpful comment. Due to the time constraint of rebuttal period, we may not be able to implement the experiments on full ImageNet, as an alternative, we have conducted experiments on Tiny ImageNet [2], a subset of ImageNet.
>
> Tiny ImageNet contains 100000 64×64 colored images of 200 classes (500 for each class), which can be regarded as more large-scale. The results are in the following Table A, also added to the revised paper. It is observed that TNA-PFN can also reduce the barriers on Tiny ImageNet by up to 23%.
>
> **Table A. Linear mode connectivity on Tiny ImageNet.** The $\rho$ for CNN is 0.4 and the $\rho$ for ResNet18 is 0.3. The learning rate is 0.08.
>
> | **Models** | **Metrics**    | **TNA-PFN**                        | **Vanilla Train** |
> |------------|----------------|------------------------------------|-------------------|
> | CNN        | Avg. Acc.      | $11.4\pm0.6$                   | $9.85\pm0.3$   |
> |            | Interp. Acc.   | $2.91\pm0.9$                   | $1.4\pm0.2$     |
> |            | Acc. Barrier   | $0.75\pm0.07$ (12.8%↓)         | $0.86\pm 0.03$  |
> |            | Loss Barrier   | $0.75\pm0.09$ (10.4%↓)         | $0.84\pm0.08$   |
> | ResNet20   | Avg. Acc.      | $31.6\pm0.4$                   | $31.8\pm0.3$    |
> |            | Interp. Acc.   | $12.5\pm2.1$                   | $6.86\pm1.8$   |
> |            | Acc. Barrier   | $0.60\pm0.07$ (23%↓)           | $0.78\pm0.06$   |
> |            | Loss Barrier   | $1.2\pm0.09$ (22.2%↓)          | $1.6\pm0.2$     |
>
>
> > 6. Response to "In the experiments in the paper, the mask ratio was set to 0.4; was this an experimentally tuned value, or is there some underlying theoretical basis?"
> >
>
> It is an experimentally appropriate value. It can be seen from Figures 1 and 11 that TNA-PFN is not sensitive in terms of $\rho$. TNA-PFN could be effetive to improve the interpolated accuracy under a wide range of $\rho$, mainly from 0.1 to 0.6. As $\rho$ is not sensitive, so we use the value of 0.4 which is moderate to balance the accuracy-connectivity tradeoff in most cases.

---

> ### Author Response · Authors · 2023-11-18
> **Response to Reviewer M3kr (3/3)**
>
> > 7. Response to "Which of the three algorithms presented in Ainsworth et al. (2023) was used for weight matching (WM)? What is the approximate computational cost per iteration?"
> >
>
> In Ainsworth et al. (2023) [3], they proposed three algorithms, activation matching, weight matching (WM), and straight-through estimators matching. The one we used is actually weight matching, which is data-independent. The computational cost of WM relies on the model's parameters, more parameters will cost more computation.
>
> > 8. Response to "Do FedPFN or FedPNU work in general learning situations other than federated learning, and can they be used for model ensembling?"
> >
>
> We appreciate the intersting comment and have conducted a preliminary experiment on model ensembling. For two indepedently trained networks, we validate the a) _Avg. Acc._: averaged accuracy of two independent models; b) _Ensemble Two_: accuracy of ensembling the two models; c) _Ensemble Four_: accuracy of ensembling the two models and two linearly interpolated models (fusion $\alpha \in \{0.3, 0.7\}$). The results are shown in the following Table B. It is found that ensembling linearly interpolated models for TNA-PFN can improve the generlization but the advantage might by marginal. We reckon it could be an interesting future research direction that studies how training-time neuron alignment improves model ensembling.
>
> **Table B. Model ensembling experiments.** The model is CNN and the dataset is CIFAR-10.
>
> |  **Metrics**    | **TNA-PFN**                        | **Vanilla Train** |
> |----------------|------------------------------------|-------------------|
> |  Avg. Acc.     | $65.10\pm0.57$         | $62.78\pm0.90$  |
> | Ensemble Two   | $70.45\pm0.12$         | $68.21\pm0.82$  |
> | Ensemble Four  | $70.53\pm0.17$         | $68.33\pm0.79$  |
>
> > 9. Response to "I'm not sure why the LoRA section was added to Appendix C.1 and what it is trying to say."
> >
>
> Thanks for the comment. It may show a potential way to verify our "subspace hypothesis" for improving training-time neuron alignment, but further investigations on the equivalent learning rates across LoRAs with different ranks are needed, so it is out of the paper's scope and we appreciate it as future work.
>
> ---
> [1] Entezari R, Sedghi H, Saukh O, et al. The Role of Permutation Invariance in Linear Mode Connectivity of Neural Networks[C]//International Conference on Learning Representations. 2022.
>
> [2] Tiny ImageNet. A subset of the ImageNet dataset. Available at: https://tiny-imagenet.herokuapp.com/.
>
> [3] Ainsworth S, Hayase J, Srinivasa S. Git Re-Basin: Merging Models modulo Permutation Symmetries[C]//The Eleventh International Conference on Learning Representations. 2023.

---

> > ### Comment · Reviewer_M3kr · 2023-11-21
> >
> > Thanks to the authors for detailed answers and additional experiments.
> > However, my concerns were not fully resolved, for the following reasons.
> >
> > 1. As mentioned in the original review, TNA-PFN is not a practical method and does not give much advantage over other LMC-related methods.
> > 2. The authors claim that the "subspace hypothesis" itself is novel in their answer, but I think it lacks novelty alone: the hypothesis itself is somewhat inferable from previous research results (Frankle, et al.), and the experiment is too simple to say that it validates it.
> >
> > I think the overall approach is impressive, and I look forward to future work with it.
> > In conclusion, I decided to keep my score.
> >
> > ---
> > **Frankle, et al.** [Linear mode connectivity and the lottery ticket hypothesis](https://arxiv.org/abs/1912.05671). _ICML_, 2020

---

> > > ### Author Response · Authors · 2023-11-21
> > > **Further Response to Reviewer M3kr**
> > >
> > > Thanks for the reviewer's feedback and the recognition of our work. We will give the responses accordingly.
> > >
> > > >  1. Response to "As mentioned in the original review, TNA-PFN is not a practical method and does not give much advantage over other LMC-related methods.".
> > > >
> > >
> > > - We note that previous LMC-related methods are _post-hoc_ while ours is orthogonal in the _training time_, which means that our method can be combined with other LMC-related algorithms for better efficiency or (sometimes) better generalization.
> > > - Additionally, our TNA-PFN-inspired federated learning algorithms, FedPFN and FedPNU, are practical and lightweight, which also lies in our algorithmic contributions.
> > >
> > > >  2. Response to "The authors claim that the "subspace hypothesis" itself is novel in their answer, but I think it lacks novelty alone: the hypothesis itself is somewhat inferable from previous research results (Frankle, et al.), and the experiment is too simple to say that it validates it.".
> > > >
> > >
> > > The work of Frankle, et al. may have different perspectives and conclusions from ours.
> > > - In Frankle, et al., the main contribution is that the authors find that better linear mode connectivity (LMC) will result in better results of lottery ticket hypothesis (LTH) _(**logical chain: LMC↑→LTH↑**)_. However, our main contribution is that the subspaces in the contexts of random pruning and partial neuron fixing will result in better linear mode connectivity _(**logical chain: random pruning & partial neuron fixing→LMC↑**)_.
> > > - Firstly, we may argue that LTH is not equivalent to random pruning & partial neuron fixing. LTH is a _pruning_ method/perspective for finding the best generalized pruned model above chance, but we study _random pruning_ and _beyond pruning—partial neuron fixing_ (TNA-PFN) without any requirements on the weight/gradient masks. Though we acknowledge that LTH may share some common patterns with our studied random pruning & partial neuron fixing, the direct causal inference of the conclusions may still not be held due to the following reasons.
> > > - Most importantly, the causal logical chains are different, making it uninferable from the previous conclusions to our current insights. Logically, "A" is the cause of "B" _(**A→B**)_ cannot directly infer to the result that "B" is the cause of "A" _(**B→A**)_. Therefore, the conclusion of Frankle, et al. that better LMC implies better LTH _(**LMC↑→LTH↑**)_ cannot directly infer to the result that LTH can result in better LMC _(**LTH→LMC↑**)_, further, cannot infer to our insights _(**random pruning & partial neuron fixing→LMC↑**)_.
> > > - Therefore, we may suppose our perspective is novel and brings some new insights and findings. We would appreciate it if the reviewer could recognize our contributions.
> > >
> > > We thank again for the reviewer's constructive comments and valuable time for the review process. Hoping our response could further resolve the concerns. We will be flattered if we receive the reviewer's positive recommendation.

---

### Official Review · Reviewer_xuvq · 2023-11-02

**Soundness:** 2 fair
**Presentation:** 2 fair
**Contribution:** 2 fair
**Rating:** 5
**Confidence:** 3

**Summary:**

In this paper, the authors introduce a subspace algorithm aimed at enhancing Linear Mode Connectivity (LMC) during the training. They begin by establishing a preliminary theorem to explore the possibility of LMC improvement through the reduction of the search space. Building on the insights gained from this theorem, they proposed a mask-based training scheme. To empirically validate the efficacy of their proposed methods, the authors conduct numerical experiments.

**Strengths:**

The paper studies an important problem that is how to improve the LCM during the training. LCM is significant in terms of training dynamics and generation and model fusion. The paper is well written and without so many typo and errors. It is easy to follow and read.

**Weaknesses:**

I have a few concerns about the novelty and results of this paper. Firstly, the authors propose a mask-based training scheme, which has already been explored in the literature. While there are variations in the masking details, such as random weight initialization for untrained weights, the overall approach resembles dropout regularization, where different subsets of neurons are masked during each iteration.

Furthermore, the results presented in the paper strike me as somewhat expected. The authors themselves acknowledge in Theorem 3.2 that improving Linear Mode Connectivity (LMC) is achieved by reducing the number of neurons, denoted by $d$. This essentially means using a simpler model, and the neural network is not necessarily overparameterized. Consequently, the number of minima is reduced, leading to a reduction in barriers between different solutions.

**Questions:**

1. Assume weight matrix $W$ are randomly initialized as IID standard Gaussian, which is a common practice in neural network initialization. Then $W$ follows a Matrix Normal distribution, i.e., $W\sim MN(0, I, I)$. By applying permutation matrices $P$ and $Q$ to $W$, the result distribution has the same Matrix Normal form, i.e., $PWQ \sim MN(0, I, I)$. Hence, the permutated weight matrix maintains the same distribution as the original $W$. This suggests that permuting the weight matrix is equivalent to reinitializing it. Given this equivalence, it raises the question of how to consider all permutations, given that the search space is uncountable.
2. The "mask ratio" is not precisely defined, and it seems to be related to the ratio of untrained neurons to the total number of neurons.
3. Is it possible to express the results in Theorem 3.2 in terms of "loss barrier" or "accuracy barrier"?

---

> ### Author Response · Authors · 2023-11-18
> **Response to Reviewer xuvq (1/3)**
>
> We appreciate the valuable reviews. We accordingly respond to the reviewer's comments below.
>
> > 1. Response to "I have a few concerns about the novelty and results of this paper. Firstly, the authors propose a mask-based training scheme, which has already been explored in the literature. While there are variations in the masking details, such as random weight initialization for untrained weights, the overall approach resembles dropout regularization, where different subsets of neurons are masked during each iteration."
> >
>
> Thanks for this comment. We would like to kindly clarify our contributions.
> - **1)** *New insights into linear mode connectivity and permutation invariance from training time.* Our novelty lies in the **training-time** "subspace hypothesis" perspective while previous works only focus on **post-training** connectivity via permutations symmetries (re-basin). The proposed TNA-PFN is just a showcase and verification of the hypothesis/perspective. The contributions are alike [1] in that its main contribution is the hypothesis of linear mode connectivity and permutation invariance and the simulated annealing algorithm (which may not be very effective) is just the verification. We believe our new perspective can inspire more future work in the community.
> - **2)** *The applications of training-time neuron alignment in federated learning.* While linear mode connectivity is somewhat theoretical, we propose two algorithms in federated learning to showcase how improving connectivity during training can help practices. The proposed methods are lightweight and effective.
>
> The mask-based methods are previously studied in the literature but none of them have been discovered to improve the linear mode connectivity during training, and that's where our new insights come from. We deem that new insights, new perspectives, and new findings can also be viewed as novelty. Also, as claimed in "Discussion on gradient/model masks" on page 4, we have elaborated on the differences between our mask and the previous masks, and it is clear that they are algorithmically different.
>
> The reviewer mentioned that our method resembles dropout, but we think this might not be accurate.
> - Dropout means pruning some weights/neurons during training, however, in Figure 2, we have shown that our TNA-PFN is different from pruning and may have better results.
> - But we need to acknowledge that TNA-PFN shares similar regularization effects with dropout/pruning, that it can improve generalization in some cases but may reduce the performances if the mask ratio is too high.
> - Another difference between dropout and our method lies in the applications of federated learning. Dropout makes the neurons dead, whereas our masking methods enable the neurons/weights to be deactivated in one round and activated in another round, which improves training-time connectivity without hurting the structure of networks.
>
>
> > 2. Response to "Furthermore, the results presented in the paper strike me as somewhat expected. The authors themselves acknowledge in Theorem 3.2 that improving Linear Mode Connectivity (LMC) is achieved by reducing the number of neurons, denoted by $d$. This essentially means using a simpler model, and the neural network is not necessarily overparameterized. Consequently, the number of minima is reduced, leading to a reduction in barriers between different solutions."
> >
>
> Thanks for this helpful comment. We realize that previous theorem may cause potential misleadings and we have refined the theorem to make it more solid.
>
> - The previous theorem may cause a misleading that lowering the network dimension by changing the network architecture can improve linear mode connectivity. Actually, this is not what we try to convey and is also inconsistent with the empirical observations.
> - After our revision, the revised theorem is more aligned with our hypothesis and method. It is proved that given a network architecture (without changing the dimension, i.e., width and depth), if we fix a proportion of parameters and train the remaining (like what TNA-PFN does), the linear mode connectivity of two models after training would be improved.
> - The revised theorem is Theorem 3.2 in blue on page 3, and the detailed proof is in the appendix. We hope the improved theorem can relieve the concerns and the potential misleading.

---

> ### Author Response · Authors · 2023-11-18
> **Response to Reviewer xuvq (2/3)**
>
> > 3. Response to "Assume weight matrix $W$ are randomly initialized as IID standard Gaussian, which is a common practice in neural network initialization. Then $W$ follows a Matrix Normal distribution, i.e., $W \sim MN(0, I, I)$. By applying permutation matrices $P$ and $Q$ to $W$, the result distribution has the same Matrix Normal form, i.e., $PWQ \sim MN(0, I, I)$. Hence, the permutated weight matrix maintains the same distribution as the original $W$. This suggests that permuting the weight matrix is equivalent to reinitializing it. Given this equivalence, it raises the question of how to consider all permutations, given that the search space is uncountable."
> >
>
> Thanks for this valuable comment. In the response, we will provide the basic background on the role of permutation invariance in linear mode connectivity to relieve the reviewer's concerns. We kindly suggest the reviewer to refer to [1]. Further, to solve the reviewer's confusion, we will provide detailed explanations from the following four aspects:
>
> - I think you may have assumed that $P$ and $Q$ are constant matrices, or $P$ and $Q$ are independent random matrices with respect to the random matrix $W$, which is $W\sim MN(0,I,I)$. Only in such cases would you arrive at the conclusion that the random matrices $W$ and $PWQ$ have the same distribution. In reality, $P$ and $Q$ are not independent random matrices with respect to $W$. In the process of using permutations to find Linear Mode Connectivity, assuming that $W$ and $U$ are the coefficient matrices of two separate models, the distribution of $P$ and $Q$ depends on $W$ and $U$. Specifically, $P, Q = \arg \min_{P,Q} |PWQ-U|$. Using such $P$ and $Q$ to permute $W$ and obtain $PWQ$ generally does not make $PWQ$ follow the same distribution as $W$. So, the understanding that "permuting the weight matrix is equivalent to reinitializing it" is incorrect. Additionally, whether $W$ and $PWQ$ follow the same distribution is not important; what matters is the distribution of $|PWQ-U|$. Even if $PWQ$ and $U$ follow the same distribution, the distribution of $|PWQ-U|$ can still be complex because $PWQ$ and $U$ are not independent. This complexity arises from their joint distribution (if you are still confused about this, please refer to https://en.wikipedia.org/wiki/Wasserstein_metric, which explains the fact that even if $A$ and $B$ follow the same distribution, the distribution of $|A-B|$ can still be complex).
>
> - A simple experiment can illustrate that $PWQ$ does not follow the same distribution as $W$. Suppose $W = \begin{pmatrix} w_{11}&w_{12}\\ w_{21}&w_{22} \end{pmatrix}$ is a 2x2 random matrix, where $w_{ij} \sim N(0,1)$. We define $Q = \begin{pmatrix}1&0\\ 0 &1\end{pmatrix}$, and $P$ follows the conditional distribution with respect to $W$: when $w_{11} \ge w_{12}$, $P = \begin{pmatrix}1&0  \\ 0&1\end{pmatrix}$, and when $w_{11} < w_{12}$, $P = \begin{pmatrix}0&1 \\ 1 &0\end{pmatrix}$. In this case, the probability density function of the element $w_{11}'$ is given by $p(x) = \frac{1}{2\pi}e^{-\frac{x^2}{2}}\int_{-\infty}^xe^{(-\frac{t^2}{2})}dt$, which does not follow the standard normal distribution (in fact, $w_{11}'$ is the order statistic of two independent normal distribution variables). So, even if $W \sim MN(0,I,I)$, it does not necessarily hold that $PWQ \sim MN(0,I,I)$. Moreover, in the context of $P,Q = \arg \min_{P,Q}|PWQ-U|$, the conditional distributions of $P$ and $Q$ with respect to $W$ and $U$ are more complex than the conditional distributions of $P_0$ and $Q_0$ in our example here. Therefore, it is not possible for $PWQ$ to follow the same distribution as $W$.
>
> - Finding when $W$ and $U$ are random matrices, the distribution of $\min|PWQ-U|$ is actually a famous question called Random Euclidean Matching Problem.  Please refer to [3].
>
> - Now, let's consider the issue of how to find $P$ and $Q$ by calculation. Indeed, for a given $W$ and $U$, finding permutations $P$ and $Q$ that satisfy $P, Q = \arg \min_{P,Q}|PWQ-U|$ is a difficult problem. It has been proven in [2] that this is an NP-hard problem. However, for a given $W$ and $U$, the search space for $P$ and $Q$ is finite, not uncountable, because $W$ and $U$ are of finite dimension and the search space for $P$ and $Q$, which is the space of permutation matrices  for $W$ and $U$ is also finite. Since finding $P$ and $Q$ is an NP-hard problem, there is no polynomial-time exact solution formula in the computation process. Typically, we use the random approximation algorithms proposed in [2], such as weight matching/activation matching.

---

> ### Author Response · Authors · 2023-11-18
> **Response to Reviewer xuvq (3/3)**
>
> > 4. Response to "The "mask ratio" is not precisely defined, and it seems to be related to the ratio of untrained neurons to the total number of neurons."
> >
>
> Thanks for this helpful comment. The reviewer understands it correctlly that the mask ratio $\rho$ refers to "keeping $\rho$ fraction of the parameters **fixed** after initialization". We have added the specific defition of $\rho$ in the revised paper (page 4, blue sentence).
>
>
> > 5. Response to "Is it possible to express the results in Theorem 3.2 in terms of "loss barrier" or "accuracy barrier"?"
> >
>
> Thank you for the suggestion. Taking your advice in the newly revised Theorem 3.2, we have reformuate $z_{\alpha}$ into the "barrier" form same with [1]. We note that the "barrier" form is depicted by the network's output function, not the loss or accuracy functions, and previous theoretical analysis also adopted this form [1].
>
> Actually, for a convex loss function $\mathcal{L}$ satisfying the $C$-Lipschitz condition, the loss barrier can be bounded by $\left|z(\alpha)\right|$: $B_{loss}(\mathbf{w_1},\mathbf{w_2})\le C\sup_\alpha|z(\alpha)|$, which differs only by the constant factor $C$, and that's why $|z(\alpha)|$ can be used for the upper bound on the barrier instead of $B_{loss}(\mathbf{w}_1,\mathbf{w}_2)$, both in our Theorem 3.2 and in the previous theoretical analysis [1].
>
> ---
> [1] Entezari R, Sedghi H, Saukh O, et al. The Role of Permutation Invariance in Linear Mode Connectivity of Neural Networks[C]//International Conference on Learning Representations. 2022.
>
> [2] Ainsworth S, Hayase J, Srinivasa S. Git Re-Basin: Merging Models modulo Permutation Symmetries[C]//The Eleventh International Conference on Learning Representations. 2023.
>
> [3] Goldman M, Trevisan D. Convergence of asymptotic costs for random Euclidean matching problems[J]. Probability and Mathematical Physics, 2021, 2(2): 341-362.

---

> ### Author Response · Authors · 2023-11-22
> **To Reviewer xuvq: hoping that our response could address your concerns**
>
> Dear Reviewer xuvq,
>
> Many thanks for your valuable comments. We have faithfully given detailed responses to your concerns during the rebuttal.
>
> Specifically, we have addressed the following points:
>
> - We have clarified our novelty and contributions and explained why the perspective and the proposed method differ from the literature.
> - We have refined our theorem to avoid misleading. The revised theorem is more solid and more aligned with the proposed TNA-PFN. Also, following your advice, the revised theorem adopts the barrier function as $z_{\alpha}$.
> - To resolve your confusion, we have detailedly explained the mechanism of the role of linear mode connectivity and permutation invariance.
> - We have corrected some minor issues, i.e., the definition of mask ratio.
>
> We would appreciate it if you could let us know if our response has sufficiently addressed your questions and thus kindly reconsider your score.
>
> Thank you.
>
> Best wishes,
>
> Authors

---

> > ### Comment · Reviewer_xuvq · 2023-11-22
> >
> > I appreciate the authors' detailed response. After reviewing all the comments and responses, I've decided to raise the score to 5 as a way to acknowledge the authors' effort in addressing some of my concerns regarding the permutation notion. However, at this point, I'm unable to suggest acceptance because I still hold the view that the masking strategy resembles dropout.
> > I maintain a different perspective from the authors regarding dropout. As the authors believe dropout's purpose is to prune weights or neurons, in my understanding, dropout spreads the utilization of every neuron by applying a fixed probability mask to each neuron during each epoch of training, ensuring diverse usage, whereas all neurons are employed during prediction.

---

> > > ### Author Response · Authors · 2023-11-23
> > > **Further Response to Reviewer xuvq**
> > >
> > > Many thanks to the reviewer's recognition of our effort during the rebuttal.
> > >
> > > We reckon that the reviewer's perspective on dropout is clear and correct. However, dropout still has essential differences with our masking strategy (i.e., TNA-PFN), therefore, causing different linear mode connectivity (LMC) between models.
> > >
> > > - __Overview.__
> > >     - Commonly, dropout randomly deactivates a proportion of _**neuron outputs**_ __in each iteration of training__, and _**the random masks change from iteration to iteration**_. While our TNA-PFN randomly generates a mask of _**neuron weights**_ __before training__ and _**apply the mask unchanged during each iteration**_ and across different models.
> > > - __Different applications of masks.__
> > >     - Firstly, dropout deactivates neurons while TNA-PFN deactivates the updates of weights (keeping the neurons activated). As shown in Figure 2 of our paper, in TNA-PFN, some weights of a neuron are fixed, but _this neuron is still **activated** through training_, whereas _dropout randomly **deactivates** some neurons_ by masking their outputs.
> > >     - Even in an extreme case where the weights of a neuron are all fixed in TNA-PFN, _this neuron is still activated_since its output still works and is passed into the next layer. It is different with dropout, where the neuron output is deactivated.
> > > - __Different randomness of masks.__
> > >     - Secondly, in dropout, _**every training iteration** has a randomly different neuron mask_, but for TNA-PFN, _the weight updating mask is randomly generated once at the beginning and **fixed through the whole training**_.
> > >     - We note that the randomness of dropout cannot improve LMC between models, because each random mask of different models for the same iteration is independent and different. Therefore, _for dropout, the models cannot be trained in a consistent subspace_, also, the potential permutation symmetries are not reduced.
> > > - __As a result, different LMCs.__
> > >     - Although dropout may share some similar effects of regularization with TNA-PFN, they have different mechanisms and different effects on LMC from our previous analysis.
> > > - __Experimental results.__
> > >     - We conduct experiments to showcase the differences between dropout and TNA-PFN. We apply dropout in each layer except for the last one. The dropout ratios are the same as the mask ratios of TNA-PFN. The results are in the following Table X.
> > >     - It can be seen that dropout cannot reduce the barriers of LMC, but TNA-PFN can. _Also, it is observed that dropout may increase the barriers._ We suppose this is probably because dropout introduces more randomness in each iteration by randomly masking neuron activations, therefore, increasing the gradient noise and causing larger LMC barriers.
> > >
> > > ---
> > >
> > > **Table X. Linear mode connectivity of dropout and TNA-PFN.** The dataset is CIFAR-10. The mask ratio/dropout ratio is $\rho$. Avg. Acc (%). refers to the averaged accuracy of two independent models. Interp. Acc. (%) refers to the accuracy of the interpolated model.
> > >
> > > | **Models ($\rho$)** | **Metrics**    | **Vanilla Train**                 |**Dropout**        |**TNA-PFN**  |
> > > |------------|----------------|------------------------------------|-------------------|-------------------|
> > > | MLP_h2_w200 ($\rho=0.2$)       | Avg. Acc.      | $44.6±0.5$                   | $38.8±0.2$   |$47.1±0.1$   |
> > > |            | Interp. Acc.   | $31.9±2$                   | $21.9±0.8$     |$38.9±1$   |
> > > |            | Acc. Barrier   | $0.285±0.05$         | $0.436±0.02$  (53%↑) |$0.174±0.02$  (38.9%↓)  |
> > > |            | Loss Barrier   | $0.299±0.06$         | $0.379±0.02$  (26.7%↑)  |$0.185±0.02$ (38.1%↓)   |
> > > | MLP_h2_w200 ($\rho=0.4$)       | Avg. Acc.    | $44.6±0.5$                   | $30.2±0.3$   |$48.3±0.4$   |
> > > |            | Interp. Acc.  | $31.9±2.4$                   | $17±0.4$     |$43.7±0.4$   |
> > > |            | Acc. Barrier  | $0.285±0.05$        | $0.441±0.02$  (54.7%↑) |$0.099±0.01$ (65.3%↓)   |
> > > |            | Loss Barrier | $0.297±0.06$        | $0.284±0.004$   (4.4%↓)  |$0.0617±0.02$  (79.2%↓)   |
> > > | CNN ($\rho=0.2$)  | Avg. Acc.      | $62.8±0.9$                   | $56.7±1$    |$65.4±0.4$   |
> > > |            | Interp. Acc.   | $47.4±3$                   | $28.8±3$   |$54.8±0.7$   |
> > > |            | Acc. Barrier   | $0.244±0.06$         | $0.493±0.04$   (102%↑)|$0.161±0.005$ (34%↓)   |
> > > |            | Loss Barrier   | $0.441±0.08$        | $0.75±0.04$ (70.1%↑)    |$0.322±0.02$  (27%↓)  |
> > > | CNN ($\rho=0.4$)  | Avg. Acc.      | $62.8±0.9$                   | $40.1±2$    |$65.1±0.6$   |
> > > |            | Interp. Acc.   | $47.4±3$                   | $20.6±1$   |$59.4±0.4$   |
> > > |            | Acc. Barrier   | $0.244±0.06$          | $0.485±0.04$   (98.8%↑)  |$0.0913±0.01$  (62.6%↓)   |
> > > |            | Loss Barrier   | $0.449±0.09$          | $0.47±0.04$   (4.7%↑)   |$0.155±0.01$  (65.5%↓)  |
> > >
> > > ---
> > > Hoping our response can further resolve your concerns. We will be flattered if we receive the reviewer's positive recommendation.

---

### Official Review · Reviewer_vDht · 2023-11-03

**Soundness:** 2 fair
**Presentation:** 2 fair
**Contribution:** 2 fair
**Rating:** 6
**Confidence:** 3

**Summary:**

The authors propose a new method for reducing symmetries in a neural network. By keeping a $1-\rho$ fraction of the parameters fixed after initialization and running gradient descent, they find a neural network that has good interpolation behavior (i.e., convex combinations of good parameters lead to parameters that are also good).

This enables federated learning and other applications, as models can be averaged without loss in performance.

**Strengths:**

- The main advantage of the method is that it is performed during training time, and is not a post-processing step. This allows for easier averaging of models.

- Extensive experiments and ablation studies.

- Applications and extensions to federated learning and interesting and possibly impactful.

**Weaknesses:**

- Some of the claims are a little overhyped, such as ``training in a subspace'' meaning keeping some set of parameters frozen and optimizing the rest.

- The theory result is weak -- it doesn't imply much for the proposed method. it states that given Gaussian initialized neural networks, the gradient and the curvature of the convex combinations of these networks are bounded by the dimension, and hence lowering the dimension is desirable.

- The algorithmic novelty is also limited, and most of the strengths lie in the empirical studies on the effect of width, etc.

**Questions:**

See weaknesses section

---

> ### Author Response · Authors · 2023-11-18
> **Response to Reviewer vDht**
>
> We appreciate the valuable reviews. We accordingly respond to the reviewer's comments below.
>
> > 1. Response to "By keeping a 1 - $\rho$ fraction of the parameters fixed after initialization ..."
> >
>
> We apologize for the unclear definition of $\rho$. Actually, the mask ratio $\rho$ refers to "keeping $\rho$ fraction of the parameters **fixed** after initialization". We have added the specific definition of $\rho$ in the revised paper (page 4, blue sentence).
>
> > 2. Response to "Some of the claims are a little overhyped, such as ``training in a subspace'' meaning keeping some set of parameters frozen and optimizing the rest."
>
> Thanks for pointing it out. We acknowledge that there are some other definitions/meanings of "subspaces" which may cause misunderstanding. To make it more rigorous and clear, we have added a footnote (page 2 in blue) to clarify the definition of "subspaces" within this paper's scope.
>
> > 3. Response to "The theory result is weak -- it doesn't imply much for the proposed method. it states that given Gaussian initialized neural networks, the gradient and the curvature of the convex combinations of these networks are bounded by the dimension, and hence lowering the dimension is desirable."
> >
>
> Thanks for this helpful comment. We realize that previous theorem may cause potential misleadings and we have refined the theorem to make it more solid.
>
> - The previous theorem may cause a misleading that lowering the network dimension by changing the network architecture can improve linear mode connectivity. Actually, this is not what we try to convey and is also inconsistent with the empirical observations.
> - After our revision, the revised theorem is more aligned with our hypothesis and method. It is proved that given a network architecture (without changing the dimension, i.e., width and depth), if we fix a proportion of parameters and train the remaining (like what TNA-PFN does), the linear mode connectivity of two models after training would be improved.
> - The revised theorem is Theorem 3.2 in blue on page 3, and the detailed proof is in the appendix. We hope the improved theorem can relieve the concerns and the potential misleading.
>
> > 4. Response to "The algorithmic novelty is also limited, and most of the strengths lie in the empirical studies on the effect of width, etc."
> >
>
> Thanks for this comment. We apologize that the previous theorem may mislead the reviewer to understand our novelty and contributions. We would like to kindly clarify our contributions.
> - **1)** *New insights into linear mode connectivity and permutation invariance from training time.* Our novelty lies in the **training-time** "subspace hypothesis" perspective while previous works only focus on **post-training** connectivity via permutations symmetries (re-basin). The proposed TNA-PFN is just a showcase and verification of the hypothesis/perspective. The contributions are alike [1] in that its main contribution is the hypothesis of linear mode connectivity and permutation invariance and the simulated annealing algorithm (which may not be very effective) is just the verification. We believe our new perspective can inspire more future work in the community.
> - **2)** *The applications of training-time neuron alignment in federated learning.* While linear mode connectivity is somewhat theoretical, we propose two algorithms in federated learning to showcase how improving connectivity during training can help practices. The proposed methods are lightweight and effective.
>
> For the width or depth issue, it is worth noting that according to the previous empirical results [1, 2], deeper networks intrinsically have higher barriers in connectivity and are therefore, harder to improve by post-training rebasin, while wider networks have lower barriers. Therefore, for our TNA-PFN, the strength of width and the weakness of depth are rational and aligned with previous understandings.
>
> However, we emphasize that our TNA-PFN has better results even on the deeper models compared with previous works. In Figure 7 of [1], the proposed simulated annealing algorithm fails to improve the post-training linear mode connectivity when the depths are high, whereas TNA-PFN improves training-time connectivity even under deeper models (Figures 3 and 4 of the paper).
>
> ---
> [1] Entezari R, Sedghi H, Saukh O, et al. The Role of Permutation Invariance in Linear Mode Connectivity of Neural Networks[C]//International Conference on Learning Representations. 2022.
>
> [2] Ainsworth S, Hayase J, Srinivasa S. Git Re-Basin: Merging Models modulo Permutation Symmetries[C]//The Eleventh International Conference on Learning Representations. 2023.

---

> ### Author Response · Authors · 2023-11-22
> **To Reviewer vDht: hoping that our response could address your concerns**
>
> Dear Reviewer vDht,
>
> Many thanks for your valuable comments. We have faithfully given detailed responses to your concerns during the rebuttal.
>
> Specifically, we have addressed the following points:
>
> - We have reformulated our claim of "subspace" to make it more rigorous.
> - We have refined our theorem to avoid misleading. The revised theorem is more solid and _more aligned with the proposed TNA-PFN_. The revised theorem proves that by keeping some weights of neurons fixed, the linear mode connectivity will be improved.
> - We have clarified our novelty and contributions and explained more about the empirical observations regarding the width and depth.
>
> We would appreciate it if you could let us know if our response has sufficiently addressed your questions and thus kindly reconsider your score.
>
> Thank you.
>
> Best wishes,
>
> Authors

---

### Public Comment · ~Seok-Ju_Hahn1 · 2023-11-11
**Requests on missing citation related to LMC-driven Federated Learning algorithm**

Dear authors,

I am the first author of SuPerFed algorithm [1] (https://dl.acm.org/doi/abs/10.1145/3534678.3539254), which proposed to induce (model-wise & layer-wise) LMC for boosting personalization performance in federated learning.

After I ran into your work, I found that my work is closely related with yours, but the citation is missing.
My work [1] has also been inspired by LMC, which is realized by orthogonalizing two endpoints (i.e., two differently initialized networks - which are a global model and a local model in my work) during backward pass in each local training time.

To the best of my knowledge, [1] was the first to introduce LMC to federated learning for a better personalization performance. In this work, we have induced LMC with more severe dataset disjointedness scenario (i.e., statistical heterogeneity in FL).

Interestingly, it was observed that the global model can be successfully connected (i.e., LMC is induced) to each different local personalized model during federated training, thus have wider minima (please see Figure A1 in p.11 of [1]) with other benefits; better personalization performances, robustness to the label noise, and broader applicability including language modeling (i.e., LMC is also induced between two differently initialized LSTM models).

Although authors of this paper have mentioned and cited some works related to (personalized) federated learning in section 5 and 6 of the paper, I think my work [1] was somewhat unfortunate to be cited.

I would like to kindly request to check these concerns and consider citing my work related to LMC-driven personalized federated learning and in your paper? I am looking forward to receiving your response.
Thank you.

Best,
Seok-Ju Hahn

Reference
[1] Hahn, S. J., Jeong, M., & Lee, J. (2022, August). Connecting Low-Loss Subspace for Personalized Federated Learning. In Proceedings of the 28th ACM SIGKDD Conference on Knowledge Discovery and Data Mining (pp. 505-515). (https://dl.acm.org/doi/abs/10.1145/3534678.3539254)

---

> ### Author Response · Authors · 2023-11-18
> **Response to Seok-Ju Hahn**
>
> Dear Seok-Ju Hahn,
>
> Thanks for pointing out your interesting work about personalized federated learning. We have added the citation in the related works part of the appendix.
>
> However, we would like to note that our work is orthogonal to personalized federated learning since our main focus is linear mode connectivity and its extensions to generalization and model fusion of federated learning. Thus, you work may seem to be out-of-scope regarding the main claim of our paper. But we still find that discussing this work can enrich the literature of federated learning, especially, it introduces linear mode connectivity to improve personalization, which is interesting and valuable, so we have cited it in the revision.
>
> Hoping our response can relieve your concern.
>
> Best wishes,
>
> The authors

---

> > ### Public Comment · ~Seok-Ju_Hahn1 · 2023-11-23
> > **Response to authors**
> >
> > Thank you very much for your positive consideration.
> > Hope you have a good final result.
> > Fingers crossed!

---

### Author Response · Authors · 2023-11-18
**General Response**

We thank the reviewers for their valuable comments and precious time. We are glad that the reviewers found our paper is *well-written and easy to follow* (Reviewers xuvq, M3kr), our *idea is novel* (Reviewer M3kr), our *research problem is important* (Reviewer xuvq), the *experiments are comprehensive* (Reviewers vDht, M3kr), and *the extensions to federated learning are interesting and practical* (Reviewers M3kr, vDht).

We find the reviewers' comments highly helpful for improving our paper, and we have incorporated them into the revised paper (**colored in blue**). To highlight, *we have refined our theorems to be more solid so as to avoid some potential misleadings, additionally, we have added experiments on large-scale dataset ---(a subset of) ImageNet.*

---
### Clarifying the contributions

We would like to kindly clarify our contributions as follows.
1. *New insights into linear mode connectivity and permutation invariance from training time.* Our novelty lies in the **training-time** "subspace hypothesis" perspective while previous works only focus on **post-training** connectivity via permutations symmetries (re-basin). The proposed TNA-PFN further showcases and verifies the hypothesis/perspective. The contributions are alike [1] that its main contribution is the hypothesis of linear mode connectivity and permutation invariance and the simulated annealing algorithm (may not be very effective) is just the verification. We believe our new perspective can inspire more future works in the community.
2. *The applications of training-time neuron alignment in federated learning.* While linear mode connectivity is somewhat theoretical, we propose two algorithms in federated learning to showcase how improving connectivity during training can help practices. The proposed methods are lightweight and effective.

---
Thanks again for the valuable comments. Specifically, we have provided detailed responses to each reviewer followed the corresponding review thread.

---
[1] Entezari R, Sedghi H, Saukh O, et al. The Role of Permutation Invariance in Linear Mode Connectivity of Neural Networks[C]//International Conference on Learning Representations. 2022.

---

### Author Response · Authors · 2023-11-21
**Hoping that our response could address your concerns**

Dear Reviewers,

Many thanks for your valuable comments. We have faithfully given detailed responses to your concerns during the rebuttal.

We would appreciate it if you could let us know if our response has sufficiently addressed your questions and thus kindly reconsider your score.

Thank you.

Best wishes,

Authors

---

### Meta-Review · Area_Chair_ewRo · 2023-12-06

**Metareview:**

The paper proposes a train-time intervention to improve linear mode connectivity (LMC). Specifically, the authors fix a subset of parameters at initiailization, which breaks some of the symmetries in the parameterization. As a result, the LMC is improved, which makes it easier to fuse models with parameter averaging. This procedure reduces the need for post-training neuron alignment such as [Git re-basin](https://arxiv.org/abs/2209.04836).

## Strengths

The paper is clear and the main contributions are intuitive. The authors also propose a practical method with promising results for federated learning. The results also add the LMC literature and provide a new perspective: train-time interventions to improve LMC.

## Weaknesses

The proposed procedure is imperfect, it doesn't fully achieve LMC (non-zero barriers). It is also not completely novel, as reviewers pointed several connections to prior work. The results are not very surprising.

Importantly, the original [mode connectivity](https://arxiv.org/abs/1802.10026) work demonstrated that any two models can be connected (trained with different random initializations, different optimizers etc). [Git re-basin](https://arxiv.org/abs/2209.04836) showed that with neural alignment we can often connect arbitrary models with LMC. From the understanding perspective, the more similar we make the two models, the more uninteresting model connectivity becomes: e.g. connecting a model to itself is trivial; connecting a model trained from the same random initialization is non-trivial, but less interesting than if the same was done for different random initializations. The authors make the models more similar by fixing some of the parameters at initialization, and show that it helps LMC, which is intuitive, but also subtle, as making the models too similar would make the result trivial.

That being said, I believe the observation is still non-trivial. Furthermore, the authors propose practical methods for federated learning based on their observation,  and achieve promising performance.

**Justification For Why Not Higher Score:**

The reviews are borderline and mixed, and identify limitations of the work. Specifically the novelty and impact of the results may be limited. No reviewer championed for the paper.

**Justification For Why Not Lower Score:**

N/A

---

### Decision · Program_Chairs · 2024-01-16

Reject